# The manganese transporter SLC39A8 links alkaline ceramidase 1 to inflammatory bowel disease

Eun-Kyung Choi[1], Thekkelnaycke M. Rajendiran[2,3], Tanu Soni[3], Jin-Ho Park[1], Luisa Aring [1], Chithra K. Muraleedharan[2], Vicky Garcia-Hernandez [2], Nobuhiko Kamada[2,4], Linda C. Samuelson[4,5], Asma Nusrat[2], Shigeki Iwase [6] & Young Ah Seo [1] ✉

The metal ion transporter SLC39A8 is associated with physiological traits and diseases, including blood manganese (Mn) levels and inflammatory bowel diseases (IBD). The mechanisms by which SLC39A8 controls Mn homeostasis and epithelial integrity remain elusive. Here, we generate *Slc39a8* intestinal epithelial cell-specific-knockout (*Slc39a8*-IEC KO) mice, which display markedly decreased Mn levels in blood and most organs. Radiotracer studies reveal impaired intestinal absorption of dietary Mn in *Slc39a8*-IEC KO mice. SLC39A8 is localized to the apical membrane and mediates $^{54}$Mn uptake in intestinal organoid monolayer cultures. Unbiased transcriptomic analysis identifies alkaline ceramidase 1 (ACER1), a key enzyme in sphingolipid metabolism, as a potential therapeutic target for SLC39A8-associated IBDs. Importantly, treatment with an ACER1 inhibitor attenuates colitis in *Slc39a8*-IEC KO mice by remedying barrier dysfunction. Our results highlight the essential roles of SLC39A8 in intestinal Mn absorption and epithelial integrity and offer a therapeutic target for IBD associated with impaired Mn homeostasis.

Inflammatory bowel diseases (IBDs), which comprise Crohn's disease and ulcerative colitis, are major chronic inflammatory disorders affecting the gastrointestinal tract in humans. The etiology and pathogenesis of IBD remain unknown; nevertheless, the prevailing notion is that a complex interplay among genetic, environmental, microbial, and immune factors contributes to the disorders[1]. A significant pathogenic factor for IBD is the disruption of the epithelial barrier function, as this leads to intestinal hyperpermeability[2]. The intestinal epithelial barrier consists of epithelial cells and intercellular junctions that seal the paracellular space and regulate the permeability of the mucosal barrier[3]. This barrier is selective and regulated, permitting the uptake of luminal nutrients while restricting the entry of

pathogens and toxins into the intestine[3]. Consequently, a compromised barrier function leads to increased intestinal permeability of luminal antigens and bacteria into the lamina propria, resulting in pathologic immune responses and chronic inflammation[4,5]. Thus, the treatment of intestinal barrier dysfunction may offer therapeutic options for IBDs.

Epidemiological studies have suggested an association between an increased risk of IBD and manganese (Mn) deficiency[6]. Mn is an essential nutrient required for fundamental physiological processes, such as bone formation, immune responses, and carbohydrate metabolism[7]. This multifunctionality likely reflects the action of this nutrient as a constituent of multiple enzymes and an activator of other

[1]Department of Nutritional Sciences, University of Michigan School of Public Health, Ann Arbor, MI, USA. [2]Department of Pathology, University of Michigan Medical School, Ann Arbor, MI, USA. [3]Michigan Regional Comprehensive Metabolomics Resource Core, University of Michigan Medical School, Ann Arbor, MI, USA. [4]Division of Gastroenterology and Hepatology, Department of Internal Medicine, University of Michigan Medical School, Ann Arbor, MI, USA. [5]Department of Molecular and Integrative Physiology, University of Michigan Medical School, Ann Arbor, MI, USA. [6]Department of Human Genetics, University of Michigan Medical School, Ann Arbor, MI, USA. ✉e-mail: youngseo@umich.edu

enzymes[8]. Mn is acquired from the diet and is abundant in plant-based foods, such as whole grains, legumes, rice, nuts, and vegetables, whereas it is deficient in animal sources, including meat, fish, poultry, eggs, and dairy products[9]. Mn deficiency has long been considered rare in humans due to its abundance in dietary sources; therefore, the argument has been made that Mn deficiency might not contribute to IBD. However, increasing evidence has indicated that dietary Mn consumption has decreased by more than 40% in the past 15 years in developed countries, including the US[9–11], likely due to increasing consumption of a Western diet characterized by high intakes of meats, sugar, and refined grains[9–11]. These studies suggest that dietary Mn deficiency is prevalent in the today's animal-based diets, paralleling the increasing incidence of IBD[12]. A causal relationship linking Mn deficiency to IBD was previously unknown; however, we have recently demonstrated that dietary Mn deficiency exacerbates the intestinal injury and inflammation associated with colitis in mice, whereas dietary Mn supplementation confers protection against colitis[13]. We also showed that dietary Mn deficiency increased intestinal permeability by impairing intestinal tight junctions[13]. This work provided evidence that Mn is necessary for proper maintenance of the intestinal barrier function and that it protects against colon injury, thereby highlighting a role for Mn in intestinal homeostasis and the occurrence of IBD[13].

In addition to dietary Mn, several traits and diseases, including whole blood Mn level[14] and Crohn's disease[15,16], are significantly associated with a common genetic variant in the metal ion transporter *SLC39A8* (p.Ala391Thr; A391T). Also known as ZIP8, SLC39A8 is a transmembrane transporter that can mediate the cellular uptake of zinc[17], iron[18], Mn[19,20], and cadmium[21]. Recent genetic studies on human subjects and animal models have revealed that SLC39A8 plays an essential role in regulating Mn levels. For example, *SLC39A8* loss-of-function mutations have been identified in patients with severe Mn deficiency[22,23]. These patients had inappropriately low blood levels of Mn, whereas the levels of other metals were normal[22,23]. We used radiotracer studies to show that SLC39A8 is a cell-surface transporter that strongly stimulates $^{54}$Mn incorporation into cells. By contrast, the disease-associated mutations completely abrogated the cellular uptake of $^{54}$Mn[24], thereby providing a causal link between SLC39A8 deficiency and Mn deficiency. Further study of hepatocyte-specific *Slc39a8* KO mice showed that SLC39A8 was localized to the apical membrane of hepatocytes, where it likely functions to reclaim Mn from the bile, suggesting that this transporter is essential for hepatic Mn homeostasis[25]. These studies establish a role for SLC39A8 in the regulation of Mn levels, further supporting the hypothesis that the SLC39A8 A391T variant may increase the susceptibility to IBD by disturbing Mn homeostasis.

SLC39A8-related diseases have only been recently discovered, and the impact of the SLC39A8 A391T variant on human health and disease is just beginning to be appreciated; therefore, our understanding of the role of SLC39A8 in Mn homeostasis and its contribution to IBD remains limited. The important unresolved issues are: (1) the role of SLC39A8 in intestinal homeostasis; (2) the mechanisms by which SLC39A8 deficiency contributes to the pathophysiology of IBD; and (3) the therapeutic targets for IBD patients with SLC39A8 deficiency. In the present study, we hypothesized that epithelial SLC39A8 controls intestinal Mn homeostasis and epithelial integrity and that dysregulation of Mn homeostasis disrupts the epithelial barrier, thereby causing a predisposition to IBD development. We tested this hypothesis by generating *Slc39a8* intestinal epithelial cell (IEC)-specific knockout (*Slc39a8*-IEC KO) mice and intestinal organoid monolayer cultures to test Mn transport and epithelial integrity. We first demonstrated that IEC-specific *Slc39a8* deletion induced Mn deficiency in the blood and multiple organs, including the intestine. We also used radiotracer $^{54}$Mn studies to show that Slc39a8 is essential for in vivo intestinal Mn absorption. We then confirmed that *Slc39a8* mediates the uptake of

$^{54}$Mn at the apical membrane of intestinal organoid monolayer cultures. Our unbiased RNA-seq analysis in the *Slc39a8*-IEC KO intestine identified alkaline ceramidase 1 (ACER1), a lipid-generating enzyme that affects intestinal permeability, as a prominent target for maintaining intestinal epithelial integrity. We then demonstrated that treatment with an ACER1 inhibitor reduced intestinal permeability by enhancing tight junction proteins, thereby suppressing intestinal injury in *Slc39a8*-IEC KO mice and intestinal organoid monolayer cultures. Our study findings establish that intestinal epithelial SLC39A8 contributes to whole-body Mn homeostasis by controlling intestinal Mn absorption and that, through this mechanism, SLC39A8 maintains intestinal epithelial integrity and protects against epithelial injury.

## Results

### Loss of intestinal epithelial *Slc39a8* leads to systemic Mn deficiency

The intestine plays a vital role in regulating Mn absorption and homeostasis in vivo[26]. *SLC39A8* is ubiquitously expressed, including in the intestine[17]. Quantitative PCR (qPCR) analysis of tissues from both female and male C57BL/6J mice revealed higher levels of *Slc39a8* in the distal small intestine and colon (Fig. 1a), with no discernible sex-specific differences. Immunofluorescent staining further showed dominant localization of SLC39A8 in the apical membrane of enterocytes in both female and male C57BL/6J mice (Fig. 1b), with no apparent sex-specific differences. The SLC39A8 staining intensity increased, with the highest expression observed in the ileum and colon. In the colon, SLC39A8 immunostaining was localized to the enterocytes lining the epithelium in the crypts. To identify the function of intestinal SLC39A8, we generated *Slc39a8* intestinal epithelial cell (IEC)-specific knockout mice (*Slc39a8*-IEC KO) by crossing floxed *Slc39a8* (*Slc39a8$^{fl/fl}$*) mice with *Villin-Cre* mice (Supplementary Fig. 1a). The qPCR analysis confirmed that *Slc39a8*-IEC KO mice had significantly decreased *Slc39a8* mRNA levels in the duodenum (–90%, $P < 0.001$), jejunum (–90%, $P < 0.001$), ileum (–93%, $P < 0.001$), and colon (–98%, $P < 0.001$), but normal *Slc39a8* expression in other tissues, including liver, heart, lung, kidney, and brain (Fig. 1c). The *Slc39a8*-IEC KO mice were born at the expected Mendelian ratios and exhibited grossly normal appearance and body weight (Supplementary Fig. 1b), indicating that IEC-specific deletion of *Slc39a8* did not lead to lethality.

We determined whether loss of *Slc39a8* in IEC alters metal homeostasis by conducting inductively coupled plasma mass spectroscopy (ICP-MS) measurements of metal concentrations in a variety of tissues from *Slc39a8*-IEC KO and control mice at 20 weeks of age. Compared to the control mice, the *Slc39a8*-IEC KO mice showed a substantial reduction in Mn concentrations not only in the ileum and colon, but also in the liver, lung, heart, and brain (Fig. 1d). Notably, whole blood Mn was markedly decreased in *Slc39a8*-IEC KO mice (Fig. 1d). No sex-specific differences were observed in Mn concentrations (Fig. 1d). Although SLC39A8 can also mediate the cellular uptake of zinc[17] and iron[18], the tissue zinc and iron levels did not differ between *Slc39a8*-IEC KO mice and control mice (Supplementary Fig. 2a, b). Loss of *Slc39a8* in the IECs also did not affect the concentrations of other metals, such as copper and selenium (Supplementary Fig. 2c, d). Thus, our *Slc39a8*-IEC KO mouse model induces a specific deletion of *Slc39a8* in the intestine and leads to systemic Mn deficiency with little impact on other metal levels.

### Slc39a8 is essential for intestinal absorption of $^{54}$Mn

The lower Mn concentrations in whole blood and other tissues of the *Slc39a8*-IEC KO mice (Fig. 1d) suggested a possible role for SLC39A8 in Mn absorption. We tested this possibility by measuring radioactivity in whole blood of control and *Slc39a8*-IEC KO mice 15 min after administration of $^{54}$Mn by oral-intragastric gavage. The

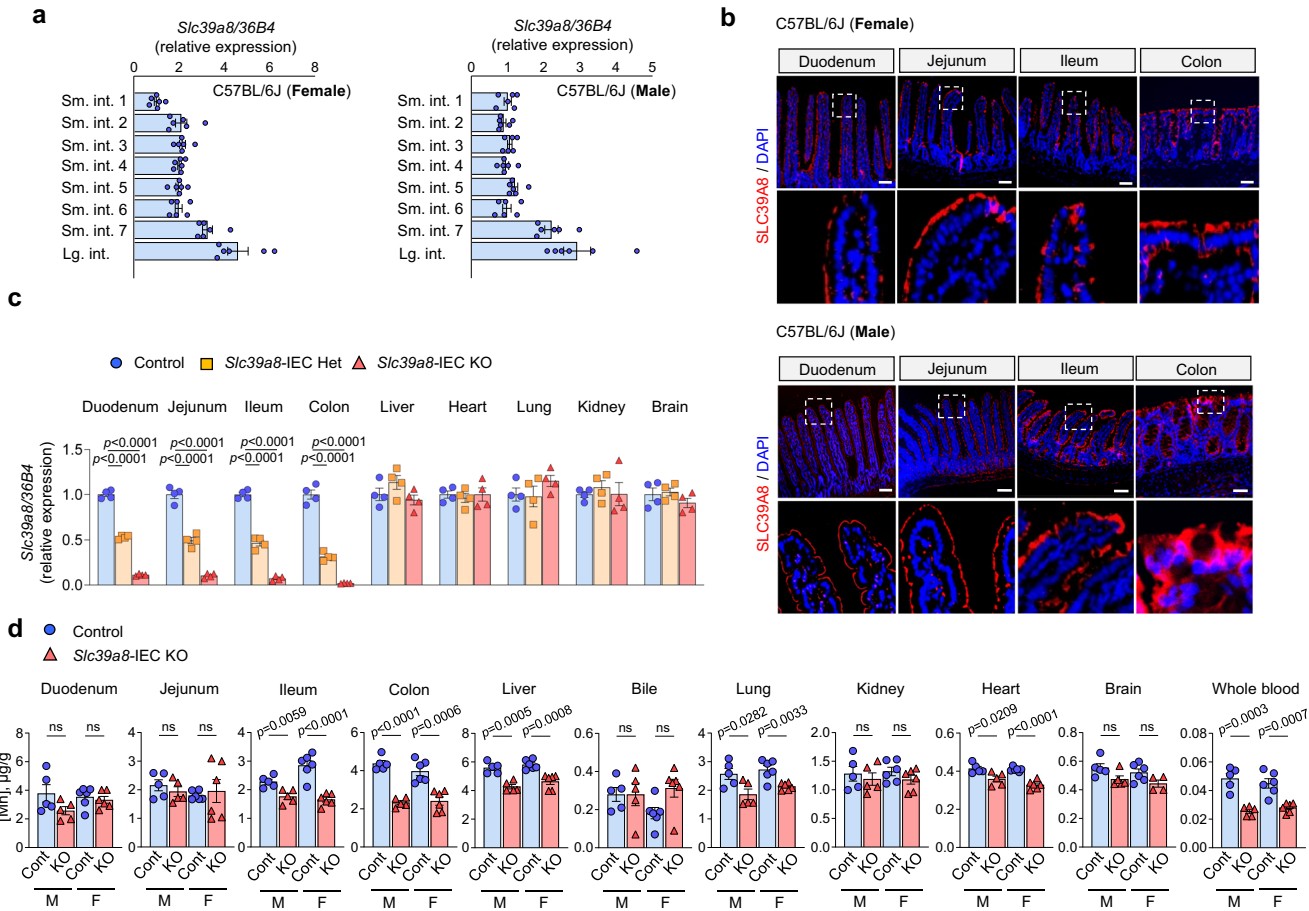

**Fig. 1 | Intestinal epithelial *Slc39a8* deletion leads to systemic Mn deficiency.**
**a** qPCR analysis of *Slc39a8* expression in 8-week-old C57BL/6J female (left) and male (right) mouse tissues (*n* = 6 per group). Small intestines were equally divided into 7 segments, and each segment is approximately 6 cm in length (Sm. int.1–7). "Small intestine 1" refers to duodenum; "Small intestine 4" refers to jejunum; "Small intestine 7" refers to ileum; and "Large intestine" refers to distal colon.
**b** Fluorescence images of frozen duodenal, jejunal, ileal, and colon sections from 8-week-old C57BL/6J female (upper) and male (lower) mice. DAPI (blue); SLC39A8 (red). Original magnification, ×10 and ×60 (enlarged insets); Scale bars: 100 µm. Each image was acquired independently three times, with similar results. **c** qPCR analysis of *Slc39a8* expression in 10-week-old male control and *Slc39a8*-IEC KO mice

(*n* = 4 per group). "Duodenum" refers to the proximal 6 cm of small intestines; "Jejunum" refers to the medial 18–24 cm of small intestines; "Ileum" refers to the medial 36–42 cm of small intestines; and "Colon" refers to the distal 4–8 cm of large intestines. **d** ICP-MS analysis of Mn levels in duodenum, jejunum, ileum, colon, liver, bile, lung, kidney, heart, brain, whole blood from 20-week-old male (M) and female (F) control (Cont) and *Slc39a8*-IEC KO (KO) mice (*n* = 5 male per group; *n* = 6 female per group). Data are presented as individual values and represent the mean ± SEM. The *p*-values were determined by one-way ANOVA with Bonferroni's multiple comparisons test for **c**, and unpaired two-tailed Student's *t* test for **d**. ns not significant. Source data are provided as a Source Data file.

15 min time point was selected based on previous studies on intestinal absorption of Mn in mice after oral-intragastric gavage[27,28]. We observed that the amount of $^{54}$Mn in the whole blood was 33% lower ($P < 0.05$) in the *Slc39a8*-IEC KO mice than in the control mice (Fig. 2a). We also measured the amount of $^{54}$Mn in the enterocytes, liver, and other tissues after administration of $^{54}$Mn via oral-intragastric gavage. Notably, the amount of $^{54}$Mn in the ileum was markedly reduced by 44% ($P < 0.05$) in *Slc39a8*-IEC KO mice compared to control mice (Fig. 2b). We also found that tissue $^{54}$Mn levels were significantly reduced in the liver (–61%, $P < 0.05$), lung (–59%, $P < 0.05$), kidney (–81%, $P < 0.05$), heart (–69%, $P < 0.05$), and femur (–41%, $P < 0.05$) in the *Slc39a8*-IEC KO mice (Fig. 2c).

The decreased $^{54}$Mn level in the blood after oral-intragastric gavage could be due to increased clearance; therefore, we measured blood clearance 15 min after intravenous injection of $^{54}$Mn. The $^{54}$Mn levels in the whole blood did not differ between the control and *Slc39a8*-IEC KO mice (Fig. 2d). The amount of $^{54}$Mn in the enterocytes, liver, and other tissues after administration of $^{54}$Mn via intravenous injection also showed no differences between control and *Slc39a8*-IEC KO mice (Fig. 2e, f). One possibility is that the

reduced $^{54}$Mn levels in the tissues after a 4 h interval from gavage or intravenous injection may reflect enhanced Mn excretion. However, we found a significant reduction in transcript levels of the intestinal Mn excretion proteins, *Slc39a14/ZIP14* and *Slc30a10/ZnT10*, in *Slc39a8*-IEC KO-derived enteroids (Supplementary Fig. 3a), thereby limiting the possibility of enhanced excretion of Mn at least in the intestine. Future studies are warranted to determine the contribution of absorption and excretion of Mn in each tissue. Overall, these data demonstrate that intestinal epithelial SLC39A8 is required for normal Mn uptake.

## Slc39a8 mediates the uptake of $^{54}$Mn at the apical membranes in intestinal organoid monolayer cultures

The major route of dietary Mn absorption is the intestine[26], where the IECs import Mn across the apical membrane into enterocytes. We examined the cellular mechanisms of Mn uptake by enterocytes by generating two-dimensional cultures of intestinal organoid monolayers directly from intestinal crypts derived from *Slc39a8*-IEC KO or control mice (Fig. 3a and Supplementary Fig. 3b)[29]. The ileum and colon were chosen as the tissues for generating the 2D enteroids

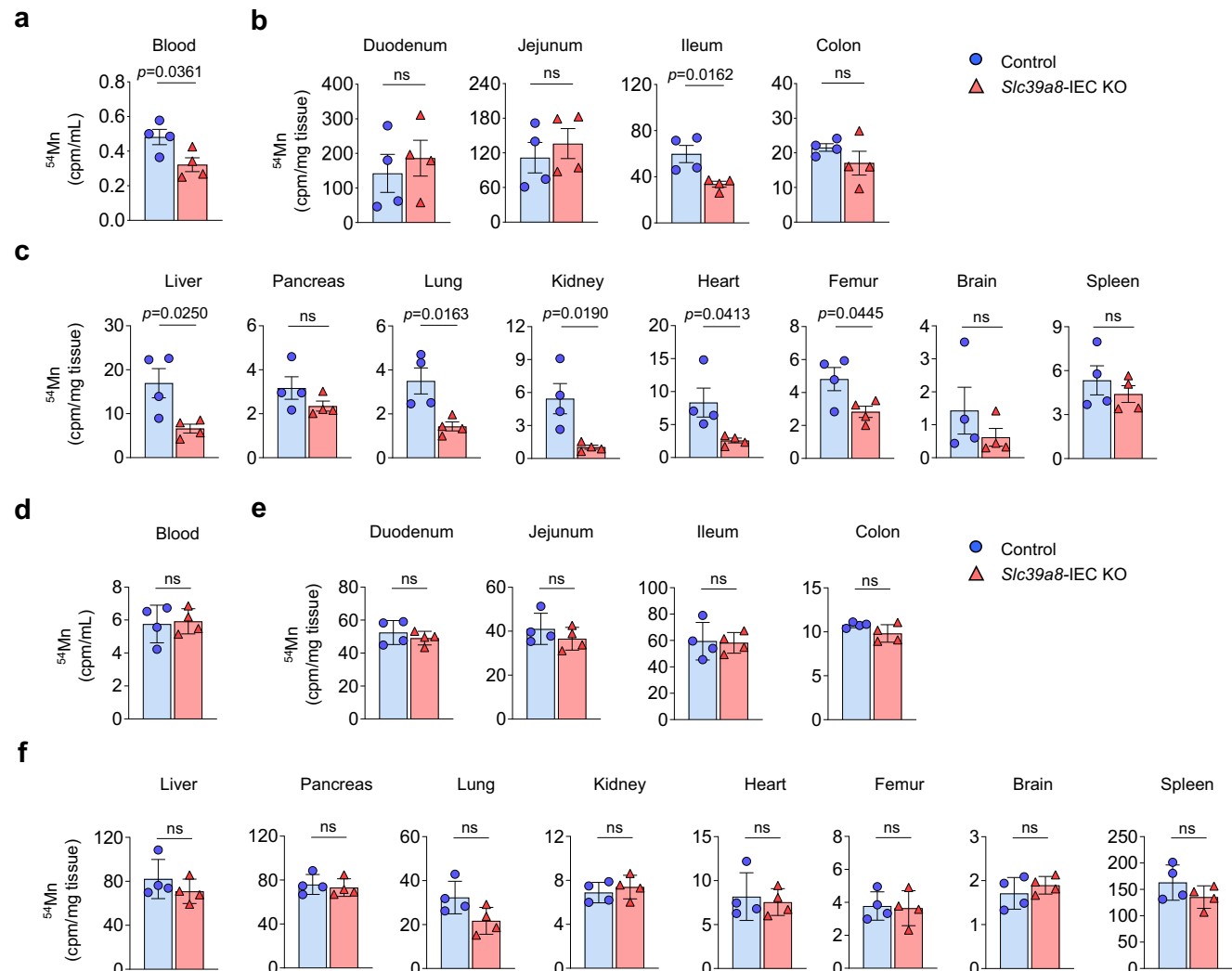

**Fig. 2 | Loss of *Slc39a8* in IECs results in impaired intestinal Mn absorption. a** Control and *Slc39a8*-IEC KO mice at 10 weeks of age were administered 0.1 μCi [54Mn]MnCl₂ per gram body weight via oral-gastric gavage. Blood was collected at 15 min, and blood counts per min (cpm) were determined by γ-counting. Mice were killed 4 h later, and (**b**) duodenum, jejunum, ileum, and colon, and (**c**) tissue cpm were determined by γ-counting. **d** Control and *Slc39a8*-IEC KO mice at 10 weeks of age were administered 0.1 μCi [54Mn]MnCl₂ per gram body weight via tail vein injection. Blood was collected at 15 min, and blood counts per minute (cpm) were determined by γ-counting. Mice were killed 4 h later, and (**e**) duodenum, jejunum, ileum, and colon, and (**f**) tissue cpm were determined by γ-counting. [control: $n = 4$, $n = 2$ male (M), $n = 2$ female (F); *Slc39a8*-IEC KO: $n = 4$ per group, $n = 2$ M, $n = 2$ F]. Data are presented as individual values and represent the mean ± SEM. The *p*-values were determined by unpaired two-tailed Student's *t* test for **a–f**. ns not significant. Source data are provided as a Source Data file.

and colonoids over other intestinal regions, as we observed greater expression of *Slc39a8* (Fig. 1a) and a larger reduction in Mn levels in those tissues (Fig. 1d). Immunofluorescence analysis confirmed the localization of SLC39A8 in control enteroid and colonoid monolayers, but its absence in *Slc39a8*-IEC KO monolayers, with an apical concentration suggested by expression next to the tight junction protein ZO-1 above the nuclear compartment (Fig. 3b). We then tested SLC39A8 Mn transport activity by adding 54Mn to the apical chamber, and measuring radioactivity in the intestinal organoid monolayer cells (Fig. 3c). Importantly, apical 54Mn accumulation was significantly impaired in the *Slc39a8*-IEC KO-derived enteroid (–35%, *P* < 0.01) and colonoid (–41%, *P* < 0.001) monolayer cells (Fig. 3d).

The residual apical 54Mn accumulation observed in *Slc39a8*-IEC KO-derived intestinal organoid monolayer cells suggests an alternative entry route for Mn at the apical membrane; therefore, we examined the expression of other metal ion transporters that could compensate Mn uptake. The *Slc39a8*-IEC KO-derived enteroids displayed significantly increased *Slc11a2/DMT1* (divalent metal

transporter 1) transcript levels and significantly decreased *Slc39a14/ ZIP14*, *Slc30a10/ZnT10*, and *Slc40a1/FPN* (ferroportin-1) transcript levels (Supplementary Fig. 3a). These data suggest that DMT1 may compensate for the loss of SLC39A8 expression in Mn uptake. We then studied the consequences of the IEC-specific deletion of *Slc39a8* on basolateral 54Mn absorption by adding 54Mn to the basolateral chamber and measuring radioactivity in the intestinal organoid monolayer cells (Fig. 3e). In contrast to the striking effect observed on apical 54Mn absorption in the *Slc39a8*-deficient IECs, 54Mn absorption from the basolateral chamber was similar in both monolayer cells (Fig. 3f). These data indicate that SLC39A8 mediates Mn uptake at the apical membrane of an intestinal organoid monolayer culture and that the loss of *Slc39a8* in IEC does not substantially affect Mn uptake from the basolateral membrane of intestinal organoid monolayer cultures. Collectively, these observations strongly support a model in which SLC39A8 is localized to the apical surface of the enterocytes, where it mediates the uptake of Mn from the intestinal lumen into enterocytes, thereby facilitating the intestinal absorption of dietary Mn.

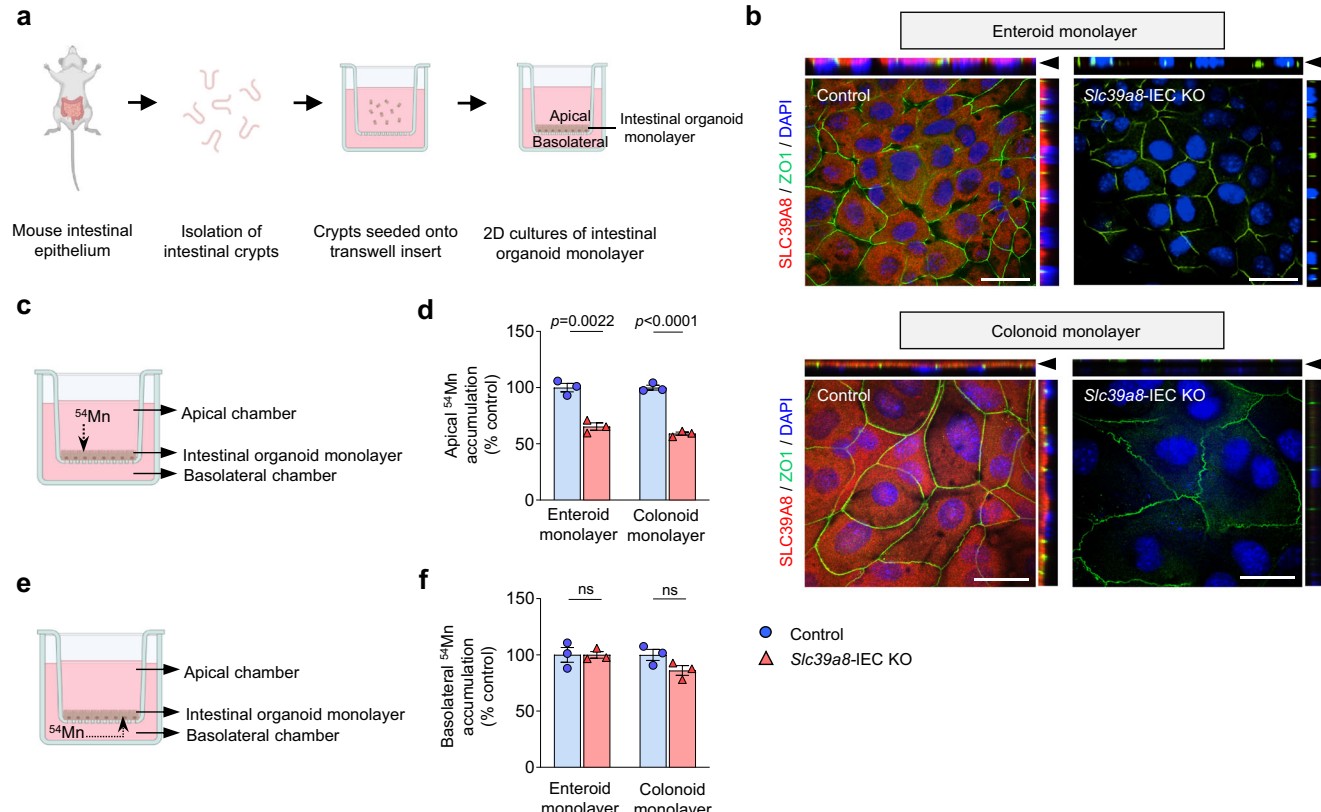

**Fig. 3 | Slc39a8 mediates the uptake of ⁵⁴Mn at the apical membrane of intestinal organoid monolayer culture. a** Schematic illustrating the generation of intestinal organoid monolayer cultures from murine intestinal crypts. **b** Representative confocal images showing immunostaining for SLC39A8 (red), ZO1 (green), and DAPI (blue) in enteroid or colonoid monolayers from control and *Slc39a8*-IEC KO mice. Polarized apical surfaces were stained with ZO1 (green), indicated by black arrowheads in the top panels. XY projection (bottom panels); XZ projection (top panels) and YZ projection (right panels) orthogonal views. Scale bar: 25 μm. Each image was acquired independently three times, with similar results. **c** Schematic representation of apical ⁵⁴Mn accumulation by intestinal organoid monolayer culture. **d** ⁵⁴Mn was added to the media in the apical

compartments of enteroid or colonoid monolayers. Cell-associated radioactivity was determined with a gamma counter (*n* = 3 biologically independent samples). **e** Schematic representation of basolateral ⁵⁴Mn accumulation by primary IECs. **f** ⁵⁴Mn was added to the media in the basolateral compartments of enteroid or colonoid monolayers. Cell-associated radioactivity was determined with a gamma counter (*n* = 3 biologically independent samples). Data are presented as individual values and represent the mean ± SEM. The *p*-values were determined by unpaired two-tailed Student's *t* test for **d** and **f**. ns not significant. Source data are provided as a Source Data file. Figures **a**, **c**, and **e** were created with BioRender.com released under a Creative Commons Attribution-NonCommercial-NoDerivs 4.0 International license.

## IEC-specific deletion of *Slc39a8* exacerbates colitis following intestinal epithelial injury

Dietary Mn deficiency has been suggested to disrupt intestinal barrier function[20]; therefore, our findings that the loss of *Slc39a8* in IEC induces intestinal Mn deficiency led us to hypothesize that the IEC-specific deletion of *Slc39a8* would alter susceptibility to colitis[30]. Dextran sodium sulfate (DSS)-induced experimental colitis is a widely used mouse model to induce acute colonic epithelial injury[30]. We treated *Slc39a8*-IEC KO and control mice with 3% DSS in the drinking water to induce colitis (inflammatory phase) and then placed on regular drinking water (recovery phase) (Fig. 4a). The DSS treatment induced colitis, characterized by rapid weight loss in control mice (Fig. 4b). Notably, *Slc39a8*-IEC KO mice displayed more pronounced body-weight loss (*P* < 0.05, Fig. 4b). Upon returning to regular water, control mice regained body weight, whereas *Slc39a8*-IEC KO mice did not show any recovery and continued to lose body weight until the experimental endpoint (Fig. 4b). As another indicator of tissue injury, the *Slc39a8*-IEC KO mice showed a dramatic decrease in colon length when compared with controls (24%, *P* < 0.001, Fig. 4c, d), indicating exacerbated DSS-induced colitis. The ratio of the spleen weight to body weight was also significantly higher in *Slc39a8*-IEC KO mice than in the control mice after DSS treatment (*P* < 0.01, Fig. 4e), indicating infiltration of splenic macrophages (Supplementary Fig. 3c)[30].

Histological analysis of the colon tissue also revealed significant epithelial degeneration, crypt destruction, severe focal ulceration, and inflammation in *Slc39a8*-IEC KO mice following DSS treatment (Fig. 4f). The total pathological score was significantly higher in the colons of *Slc39a8*-IEC KO mice than in control mice (Fig. 4g). Consistent with previous studies[31,32], we observed lower sensitivity of female *Slc39a8*-IEC KO mice to the effects of DSS-induced colitis (Supplementary Fig. 3d). Since Slc39a8 expression and Mn levels did not differ between sexes (Figs. 1 and 2), the cause of the lower sensitivity of females to DSS is unknown. Based on these observations, we focused on our further study on male mice.

We gained further insight into the change in intestinal permeability after DSS injury by examining barrier function using FITC-dextran (4 kDa molecular weight) as an indicator. Serum FITC-dextran levels on day 13 of DSS treatment were significantly higher in *Slc39a8*-IEC KO mice than in control mice (Fig. 4h). Increased permeability could be attributed to mucosal erosion and impaired epithelial barrier function[33]. Therefore, we examined the expression of tight junction proteins, as these proteins can directly affect intestinal permeability. In *Slc39a8*-IEC KO mice, our qPCR analysis showed significantly greater reductions in the expression levels of tight junction genes *Cldn2, Cldn3, Cldn5,* and *Cldn7* in the colonic mucosa after DSS treatment compared to control mice (Fig. 4i).

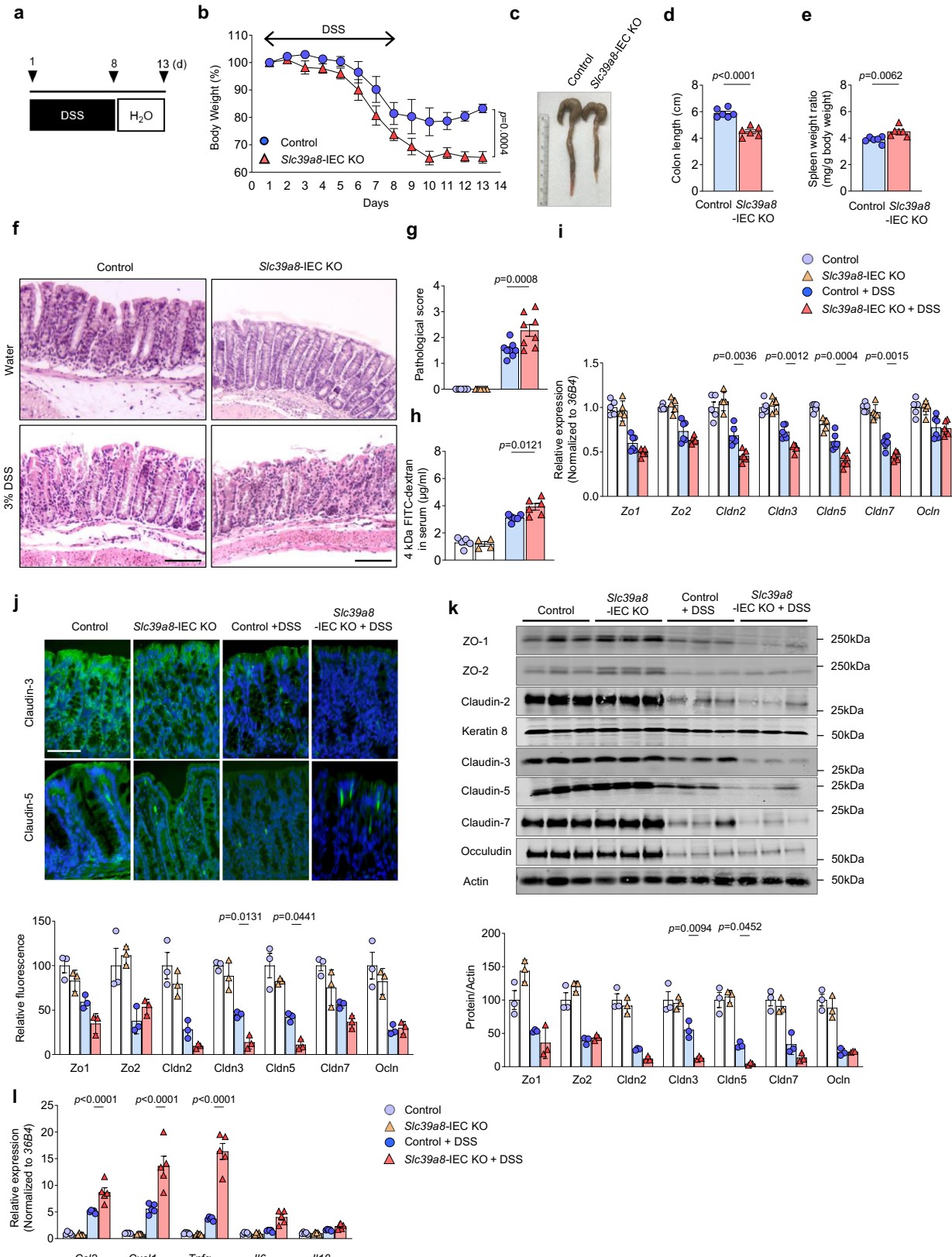

Immunofluorescence staining revealed that Zo1, Zo2, Cldn2, Cldn5, and Ocln were localized at the apical side of the colonic epithelium, while Cldn3 and Cldn7 were detected at the lateral membrane (Fig. 4j and Supplementary Fig. 3e). DSS treatment severely impaired the localization and expression of these tight junction proteins (Fig. 4j and Supplementary Fig. 3e). Notably, significantly reduced expression

levels of Cldn3 and Cldn5 were observed in the *Slc39a8*-IEC KO mice after DSS treatment (Fig. 4j). Immunoblotting analysis confirmed diminished levels of Cldn3 and Cldn5 (Fig. 4k). While Cldn2 levels can increase in the inflammatory state in human tissues[34,35], our data and findings by others[36,37] have indicated decreases in Cldn2 levels (Fig. 4k and Supplementary Fig. 3f), implicating a complex context-dependent

**Fig. 4 | Loss of *Slc39a8* in IECs exacerbates DSS-induced acute model of colitis.**
**a** Schematic of the DSS-induced acute colitis model. For colitis induction, mice were given drinking water containing 3% (w/v) DSS for 8 days (inflammatory phase). The mice were then provided with regular drinking water for 5 days (recovery phase). Representative data from two independent experiments are shown.
**b** Changes in body weight (percentage of original body weight) over time (days) in 8-week-old male control and *Slc39a8*-IEC KO mice following DSS treatment ($n = 13$ per group). **c** Gross morphology of the colon. **d** Colon length and **e** spleen/body weight ratios on day 13 after the DSS treatment in control and *Slc39a8*-IEC KO mice ($n = 6$ per group). **f** Hematoxylin/eosin (H&E) staining of colons and **g** pathological scores on day 13 without or with DSS treatment in control and *Slc39a8*-IEC KO mice ($n = 8$ per group). Scale bars: 100 μm. **h** Measurement of 4 kDa FITC-dextran in serum on day 13 without or with the DSS treatment in control and *Slc39a8*-IEC KO mice ($n = 6$ per group). **i** qPCR quantification (control, $n = 5$; *Slc39a8*-IEC KO, $n = 5$; control + DSS, $n = 6$; *Slc39a8*-IEC KO + DSS, $n = 6$), **j** immunofluorescence staining and relative fluorescence, and **k** immunoblot analysis and densitometric quantification of tight junction proteins in the colons of control and *Slc39a8*-IEC KO mice on day 13 without or with DSS treatment ($n = 3$ per group). Scale bar: 100 μm.
**l** qPCR quantification of proinflammatory cytokines and chemokines in the colons of control and *Slc39a8*-IEC KO mice on day 13 without or with DSS treatment ($n = 5$ per group). Data are presented as individual values and represent the mean ± SEM. The *p*-values were determined by two-way ANOVA with Bonferroni's multiple comparisons test for **b**, unpaired two-tailed Student's *t* test for **d** and **e**, and one-way ANOVA with Bonferroni's multiple comparisons test for **g–l**. Source data are provided as a Source Data file.

Cldn2 regulation. These data suggest a selective impact of Slc39a8 on the transcripts and proteins related to tight junctions. Compared to control mice, *Slc39a8*-IEC KO mice also showed significantly higher expression of the chemokine genes *Ccl2* and *Cxcl1* and the inflammatory cytokine gene *Tnfa* (Fig. 4l). No significant differences were observed between control and *Slc39a8*-IEC KO mice in the measured parameters without DSS treatment, (Fig. 4f–l).

Exacerbated colitis was also examined in *Slc39a8*-IEC KO mice using a second model of 2,4,6-trinitrobenzene sulfonic acid (TNBS)-induced colitis. Intrarectal administration of the haptenating agent TNBS induces a severe CD4[+] T cell-dependent colitis in mice, resulting in increased permeability and inflammation[38]. Our results were similar for both the TNBS-induced colitis model and the DSS-induced colitis model. *Slc39a8*-IEC KO mice exhibited exacerbated TNBS-induced body weight loss, clinical scores, and colonic injury compared to control mice (Supplementary Fig. 4a–e). Additionally, *Slc39a8*-IEC KO mice displayed increased gut permeability, reduced expression of tight junction protein transcripts of *Zo1* and *Cldn5*, and increased expression of the chemokine genes *Ccl2* and *Cxcl1*, and the inflammatory cytokine gene *Tnfa* (Supplementary Fig. 4f–h). These findings demonstrate that IEC-targeted *Slc39a8* deficiency exacerbates mucosal damage by disrupting barrier function in both DSS and TNBS colitis models.

We further verified the colitis phenotype of *Slc39a8*-IEC KO mice in a third model of chronic DSS colitis that employs cyclical administration of DSS in drinking water followed by recovery with water alone. Mice received two cycles of 3% DSS treatment, each cycle consisting of 8 days of the inflammatory phase followed by 5 days of recovery phase. In the second DSS cycle, the clinical scores were assessed daily[39] (Fig. 5a). The *Slc39a8*-IEC KO mice developed clinical symptoms of colitis with increasing severity starting on day 2 of the second cycle and peaking on day 11, whereas the symptoms in control mice were delayed and significantly reduced in severity (Fig. 5b). The *Slc39a8*-IEC KO mice also showed an approximately 29% decrease in colon length compared with the control mice (Fig. 5c). Moreover, approximately 50% of the *Slc39a8*-IEC KO mice died by day 13 of the second cycle (Fig. 5d). Histological examination confirmed a marked increase in immune cell infiltration and ulceration in the *Slc39a8*-IEC KO colon (Fig. 5e), with significantly higher pathological scores on day 13 of the second DSS cycle (Fig. 5f). Without DSS treatment, we observed no significant differences between control and *Slc39a8*-IEC KO mice in the measured parameters (Fig. 5c–f). Taken together, the data from both the acute DSS colitis and TNBS colitis model, as well as the chronic DSS colitis model, indicated that the loss of epithelial Slc39a8 exacerbates mucosal barrier damage and promotes colon inflammation and colitis.

## Alkaline ceramidase 1 is upregulated in the intestines of *Slc39a8*-IEC KO mice

To identify the mechanisms by which IEC-specific *Slc39a8* deficiency impairs intestinal integrity and exacerbates intestinal injury, we employed an unbiased RNA-seq approach using ileal and colonic mucosa from *Slc39a8*-IEC KO and control mice. Differential gene expression analyses identified four genes with consistently altered expression in the *Slc39a8*-IEC KO intestines compared to the control (*Padj* < 0.1, ileum and colon data were combined, Fig. 6a and Supplementary Table 1). Of the four genes, *Slc39a8* expression was downregulated in the *Slc39a8*-IEC KO intestines, demonstrating the validity of the RNA-seq approach. Among the three upregulated genes, alkaline ceramidase 1 (*Acer1*) drew our attention (Fig. 6a, b). ACER1 is localized to the endoplasmic reticulum (ER)-Golgi network and catalyzes the hydrolysis of ceramides into sphingosine and free fatty acids at an alkaline pH 7.5–9.5[40]. Therefore, ACER1 is a key enzyme in sphingolipid metabolism, a process that is involved in mouse models of colitis and human IBD[41–43]. ACER1 is also critical for regulating ceramide content[44]. Our qPCR analysis using an independent set of RNA samples confirmed the upregulation of *Acer1* expression in the ileal and colonic mucosa from *Slc39a8*-IEC KO mice compared to control mice (Fig. 6c). The upregulation of *Acer1* expression was also confirmed in the enteroid and colonoid monolayers derived from control and *Slc39a8*-IEC KO mice (Fig. 6d). Further analysis by immunoblotting confirmed an increase in Acer1 protein levels in the ileum (-1.4 fold, *P* < 0.05) and colon (-1.3 fold, *P* < 0.05) of *Slc39a8*-IEC KO mice and the enteroid monolayers (~4.5 fold) of *Slc39a8*-IEC KO mice (Fig. 6e, f). Although the increases in Acer1 protein levels were relatively smaller in the colonoid monolayers (1.3 fold) of *Slc39a8*-IEC KO mice, these smaller changes in the colon were consistent with the RNA-seq results. These results demonstrate that Acer1 is upregulated in the intestines of *Slc39a8*-IEC KO mice.

## Acer1 inhibitor enhances intestinal barrier function in *Slc39a8*-IEC KO mice-derived intestinal organoid monolayer culture

The finding of higher expression levels of *Acer1* in the ileum and colon (Fig. 6) led us to test the contribution of ACER1 to the impaired intestinal barrier function and IBD susceptibility of *Slc39a8*-IEC KO mice. We selected the ACER1 inhibitor (1S,2R)-d-*erythro*-2-(*N*-myristoylamino)−1-phenyl-1-propanol (D-*e*-MAPP) as it potently and specifically inhibits alkaline ceramidase activity but not acid or neutral ceramidase activity[45]. We first examined the effect of D-*e*-MAPP on epithelial barrier function in intestinal organoid monolayer cultures derived from *Slc39a8*-IEC KO and control mice under normal conditions. D-*e*-MAPP significantly upregulated the tight junction protein transcripts of *Zo1*, *Cldn3*, *Cldn5*, and *Cldn7* in *Slc39a8*-IEC KO mice-derived enteroid and colonoid monolayers (Fig. 7a–d). Further analysis by immunoblotting confirmed that D-*e*-MAPP treatment enhanced levels of tight junction proteins Zo1, Cldn5, and Cldn7 in enteroid monolayers derived from *Slc39a8*-IEC KO mice (Fig. 7e). A similar trend was observed for the Cldn3 protein levels, although it did not reach statistical significance. We then performed FITC-dextran permeability assays and transepithelial electrical resistance (TEER) measurements. D-*e*-MAPP treatment significantly inhibited leakage of FITC-dextran into the bottom chambers in *Slc39a8*-IEC KO mice-derived enteroid

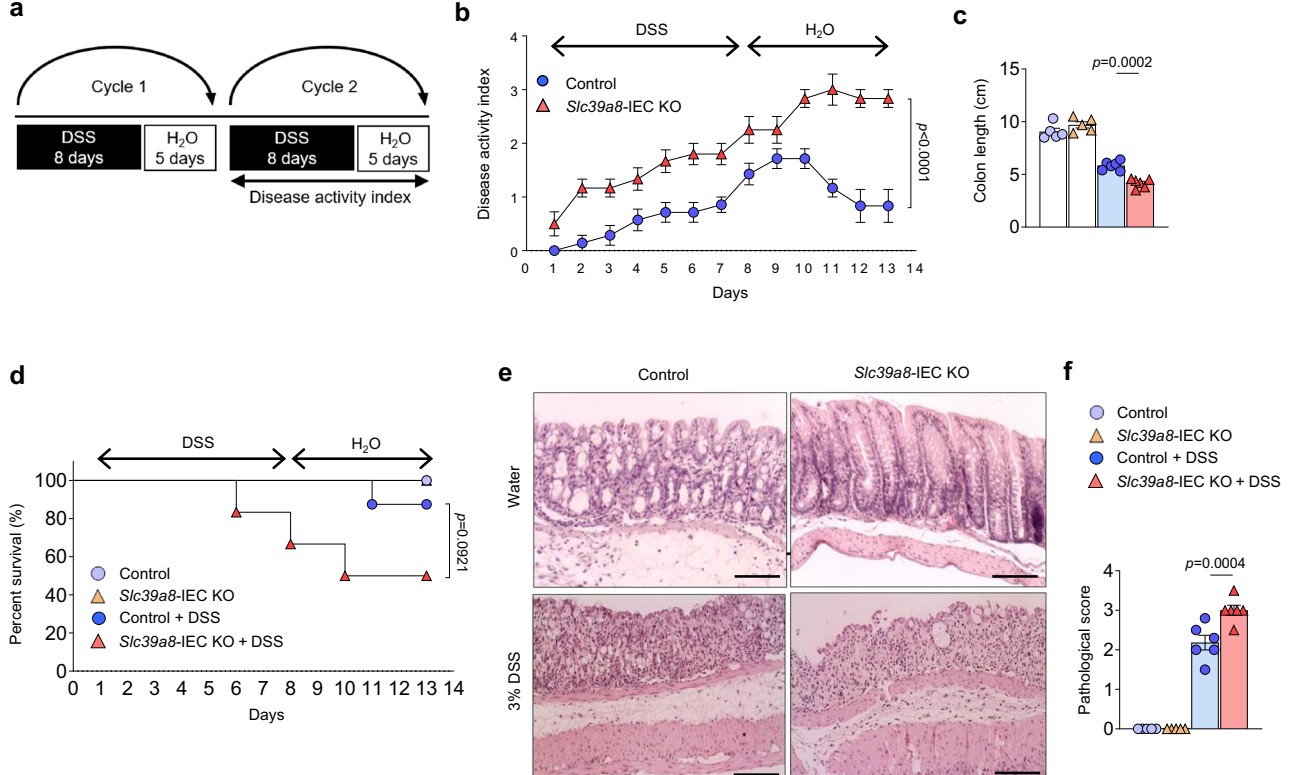

**Fig. 5 | Loss of epithelial Slc39a8 aggravates the DSS-induced chronic model of colitis. a** Schematic of the DSS-induced chronic colitis model. For colitis induction, mice received two cycles of DSS treatment provided in drinking water. Each cycle consisted of 8 days of water containing 3% DSS (inflammatory phase) followed by 5 days of water alone (recovery phase). Representative data from two independent experiments are shown. **b** Time course of the disease activity index in control and *Slc39a8*-IEC KO mice during the second DSS cycle (*n* = 6 per group). **c** Colon length from control and *Slc39a8*-IEC KO mice on day 14 without or with DSS treatment (control, *n* = 5; *Slc39a8*-IEC KO, *n* = 5; control + DSS, *n* = 6; *Slc39a8*-IEC KO + DSS, *n* = 6 per group). **d** Survival percentage (%) over time. **e** H&E staining of colons and **f** pathological scores from control and *Slc39a8*-IEC KO mice on day 14 without or with DSS treatment (control, *n* = 5; *Slc39a8*-IEC KO, *n* = 5; control + DSS, *n* = 6; *Slc39a8*-IEC KO + DSS, *n* = 6 per group). Scale bars: 100 μm. Data are presented as individual values and represent the mean ± SEM. The *p*-values were determined by two-way ANOVA with Bonferroni's multiple comparisons test for **b**, one-way ANOVA with Bonferroni's multiple comparisons test for **c** and **f**, and log-rank test for **d**. Source data are provided as a Source Data file.

and colonoid monolayers (Fig. 7f). Notably, at the highest D-*e*-MAPP dose tested, intestinal permeability, as determined by 4-kDa FITC-dextran, was increased to control-like levels, but no higher (Fig. 7f). Furthermore, the epithelial barrier function analysis by TEER measurements revealed that D-*e*-MAPP significantly increased TEER in the enteroid and colonoid monolayers derived from *Slc39a8*-IEC KO mice (Fig. 7g). The D-*e*-MAPP treatment restored the epithelial barrier function determined by TEER measurements back to control-like levels, but no lower (Fig. 7g). Treatment of intestinal organoid monolayer cultures with D-*e*-MAPP at the indicated doses also did not cause any observable toxicity (Supplementary Fig. 5a). Overall, these results suggest that ACER1 inhibition restores the gut barrier integrity in *Slc39a8*-deficient enterocytes by increasing the expression of tight junction proteins under normal conditions.

### Treatment with Acer1 inhibitor D-*e*-MAPP mitigates colitis in *Slc39a8*-IEC KO mice

Preventing increases in gut permeability by enhancing barrier functions could represent a potential therapeutic avenue for treating IBD patients with *SLC39A8* deficiency. We hypothesized that targeting ACER1 may be critical for maintaining epithelial integrity and barrier function to protect against intestinal injury and inflammation in *Slc39a8*-IEC KO mice. We therefore examined the physiological relevance of ACER1 inhibition in the DSS-induced colitis model. The ACER1 inhibitor exhibited strong barrier protective activities by upregulating tight junction proteins in *Slc39a8*-deficient intestinal organoid

monolayer culture under normal conditions (without DSS challenge) (Fig. 7). We therefore investigated whether regular exposure to this inhibitor would have sustained and beneficial colitis-preventing effects. Mice were pretreated with or without D-*e*-MAPP via oral gavage administration (10 nmol/g at 24 h intervals) for one week, followed by DSS treatment to induce colitis (inflammatory phase) (Fig. 8a). The mice were then placed on regular drinking water (recovery phase). The *Slc39a8*-IEC KO mice pretreated with D-*e*-MAPP were significantly protected from DSS-induced loss of body weight (Fig. 8b) and colon shortening (Fig. 8c, d). The pretreated *Slc39a8*-IEC KO mice also showed significantly reduced inflammation and pathological scores compared to untreated *Slc39a8*-IEC KO mice (Fig. 8e, f). Notably, we observed reduced intestinal permeability in the *Slc39a8*-IEC KO mice pretreated with D-*e*-MAPP (Fig. 8g). Further analysis of tight junction proteins in these mice also showed that pretreatment with D-*e*-MAPP protected against the downregulation of *Zo1* and *Cldn3* (Fig. 8h). Pretreatment with D-*e*-MAPP in *Slc39a8*-IEC KO mice also reduced the expression of inflammation markers, including *Cxcl1* and *Tnfa*, which are hallmarks of ulcerative colitis (Fig. 8i). Pretreatment of *Slc39a8*-IEC KO or control mice with D-*e*-MAPP did not cause any signs of toxicity, as no changes were observed in mouse body weights or serum AST and ALT levels (Supplementary Fig. 5b–d).

The therapeutic applications of D-*e*-MAPP were also examined in the DSS-induced colitis model (Supplementary Fig. 6a). This treatment significantly protected the *Slc39a8*-IEC KO mice from DSS-induced body weight loss (Supplementary Fig. 6b) and colon shortening

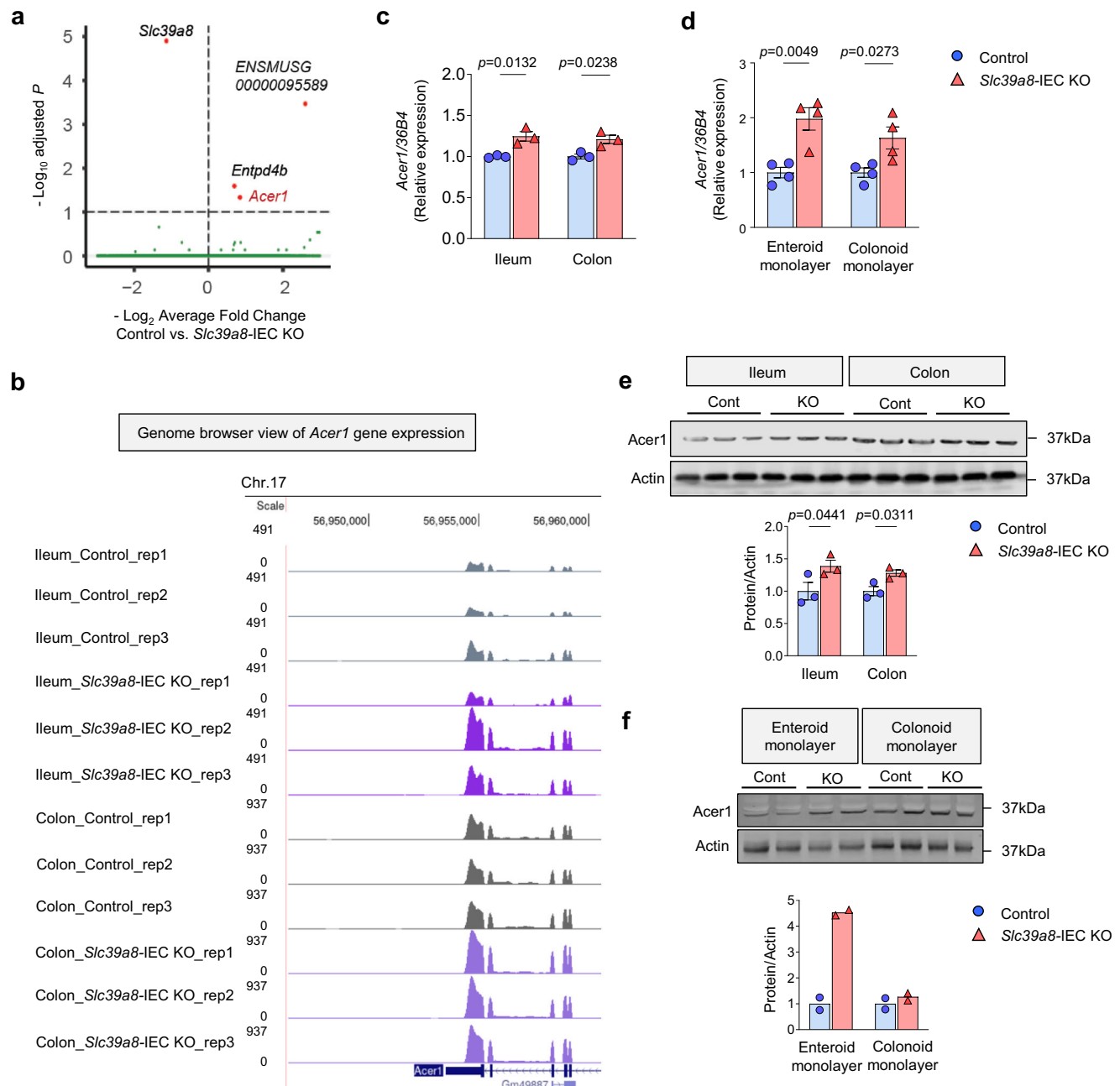

**Fig. 6 | ACER1 is upregulated in the intestine of *Slc39a8*-IEC KO mice. a** RNA-seq analysis revealed 4 misregulated genes (*P*adj < 0.1, *n* = 3) in the intestines of 8-week-old male *Slc39a8*-IEC KO mice. Volcano plot profiles of the -log10 adjusted *p*-value and log2 fold change of gene expression between control and *Slc39a8*-IEC KO mice intestines. **b** Genome browser shot of the *Acer1* locus. Validation of *Acer1* expression by qPCR in **c** the ileal and colonic mucosa in control and *Slc39a8*-IEC KO mice (*n* = 3 per group), and **d** the enteroid and colonoid monolayers derived from control or *Slc39a8*-IEC KO mice (*n* = 4 biologically independent samples per group). Validation of Acer1 protein expression by immunoblot analysis in **e** the ileal and colonic mucosa in control and *Slc39a8*-IEC KO mice (*n* = 3 per group), and **f** the enteroid and colonoid monolayers derived from control and *Slc39a8*-IEC KO mice (*n* = 2 biologically independent samples). Immunoblots of actin were used as the loading control. Quantification of the relative protein expression after normalization with actin. Data are presented as individual values and represent the mean ± SEM. The *p*-values were determined by Wald test implemented in DESeq2 R package for **a**, and unpaired two-tailed Student's *t* test for **c**–**e**. Source data are provided as a Source Data file.

(Supplementary Fig. 6c, d). Histological analysis also showed significantly less tissue damage and inflammation scores in *Slc39a8*-IEC KO mice treated with D-*e*-MAPP (Supplementary Fig. 6e, f). The intestinal permeability was reduced in *Slc39a8*-IEC KO mice treated with D-*e*-MAPP compared to those without D-*e*-MAPP treatment, although this trend did not reach statistical significance (Supplementary Fig. 6g). Furthermore, D-*e*-MAPP treatment provided protection against DSS-induced downregulation of *Cldn3* in the colon of *Slc39a8*-

IEC KO mice (Supplementary Fig. 6h) and reduced the expression of inflammatory markers, such as *Tnfa* and *Il1b* (Supplementary Fig. 6i). These results suggest that the ACER1 inhibitor has beneficial effects in preserving the barrier integrity and mitigating colitis in *Slc39a8*-IEC KO mice.

To determine the mechanisms underlying the upregulation of ACER1 and the rescue effect of D-*e*-MAPP, we performed global, unbiased lipidomics profiling of the intestine from WT and mutant

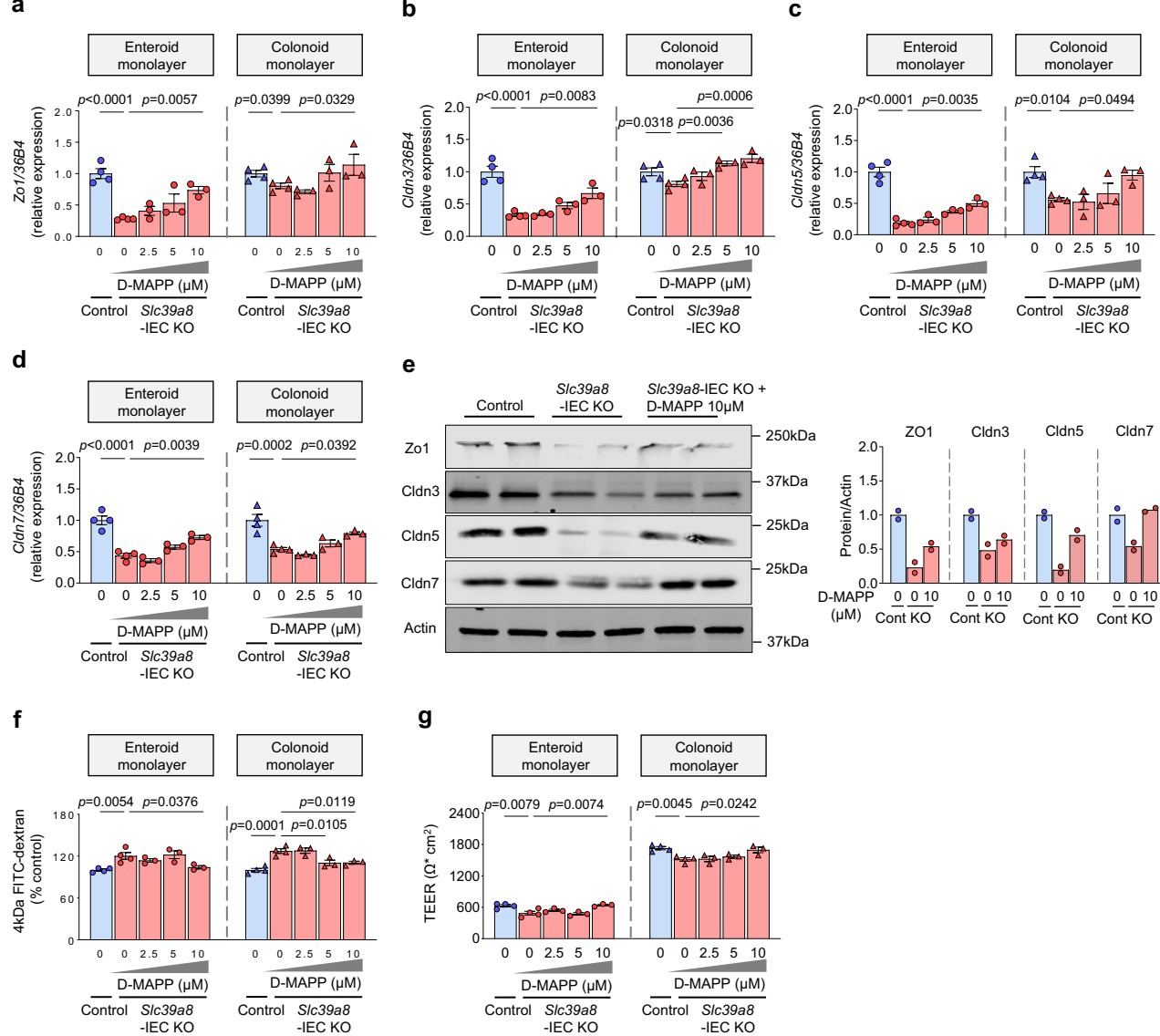

**Fig. 7 | ACER1 inhibitor D-*e*-MAPP promotes upregulation of epithelial tight junction and epithelial barrier function in *Slc39a8*-deficient intestinal organoid monolayer cultures.** qPCR analysis of tight junction (**a**) *Zo1*, (**b**) *Cldn3*, (**c**) *Cldn5*, and (**d**) *Cldn7* measured 24 h after D-*e*-MAPP treatment in *Slc39a8*-IEC KO mouse-derived enteroid or colonoid monolayers (Without D-*e*-MAPP treatment, n = 4; D-*e*-MAPP treatment, n = 3 biologically independent samples). **e** Immunoblot analysis and densitometric quantification of ZO1, Cldn3, Cldn5, and Cldn7 measured 24 h after D-*e*-MAPP treatment in *Slc39a8*-IEC KO mouse-derived enteroid monolayers. Quantification of the relative protein expression after normalization with actin (n = 2 biologically independent samples). **f** Paracellular flux of 4 kDa FITC-dextran and **g** Intestinal transepithelial electrical resistance (TEER) were measured 24 h after D-*e*-MAPP treatment in *Slc39a8*-IEC KO mouse-derived enteroid and colonoid monolayers (Without D-*e*-MAPP treatment, n = 4; D-*e*-MAPP treatment, n = 3 biologically independent samples). Data are presented as individual values and represent the mean ± SEM. The *p*-values were determined by one-way ANOVA with Bonferroni's multiple comparisons test for **a**–**d**, **f**, and **g**. Source data are provided as a Source Data file.

mice, with or without the inhibitor, using triple time of flight liquid chromatography-mass spectrometry (Triple-TOF LC-MS). While ACER1 converts ceramide into sphingosine, sphingomyelin synthase (SMS1) converts ceramide into sphingomyelin; therefore, the two enzymes represent two major arms of ceramide metabolism (Fig. 8j). The lipidome profiling identified 40 differentially regulated lipids in *Slc39a8*-IEC KO intestines compared to control (*P*adj < 0.2, Supplementary Fig. 7a). Among the altered lipids, the heatmap (Fig. 8k) represents lipid species relevant to ceramide metabolism, including sphingomyelins, hexosylceramides, sphinganine, phosphatidylethanolamine ceramides, and ceramides. While we predicted a reduction in ceramides following ACER1 upregulation, the reduction was only observed for Cer 38:2;2O and Cer 43:1;2O, whereas

four other ceramide species were upregulated in the *Slc39a8*-IEC KO intestines.

Notably, sphingomyelins were all significantly downregulated in the mutants, which is consistent with the fact that Mn is necessary for the activity of ceramide phosphoethanolamine synthase (Cpes)[46], the insect homolog of SMS1[47]. Indeed, SMS activity was lower in *Slc39a8*-IEC KO intestines compared to control intestines (Fig. 8l). Furthermore, all three sphingomyelin species showed partial or complete restoration of their abundance upon D-*e*-MAPP treatment; the inhibitor alone did not affect the sphingomyelin levels (Fig. 8m). In contrast, we did not find any ceramide species whose changes completely agreed with the phenotypic outcomes, i.e., dysregulation in *Slc39a8*-IEC KO, restoration by D-*e*-MAPP, and no change by inhibitor alone

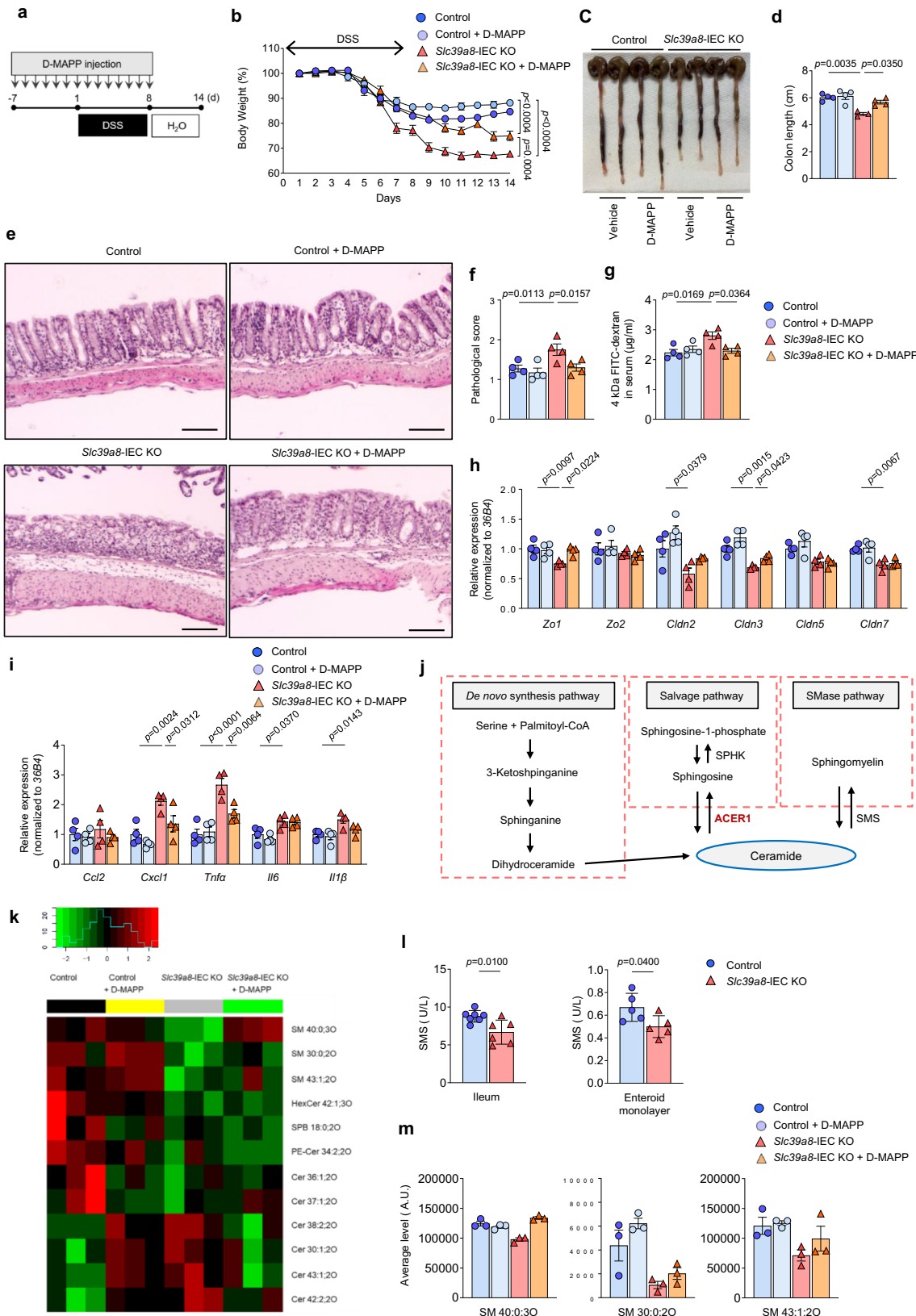

(Supplementary Fig. 7b–g); the six KO-dysregulated ceramides largely satisfied the first two criteria, but D-*e*-MAPP alone also changed their levels in the control intestine, ruling them out as the mediator for gut phenotype rescue effects.

These results led us to propose the following mechanisms underlying Acer1 upregulation in *Slc39a8*-IEC KO intestines and

phenotypic rescues by D-*e*-MAPP. First, the reduced SMS1 activity due to Mn deficiency led to reduced sphingomyelins and increased ceramide, triggering the compensatory increase of Acer1 expression to offset the ceramide increases. The increased Acer1 expression indirectly led to alterations in lipid composition, including the upregulation of some ceramide species and dysregulation of other detected lipids.

**Fig. 8 | Pretreatment with the ACER1 inhibitor D-e-MAPP confers protection against colitis in *Slc39a8*-IEC KO mice. a** Schematic of pretreatment with the ACER1 inhibitor D-e-MAPP in the DSS-induced colitis model in mice. Mice were treated with or without the ACER1 inhibitor D-e-MAPP (10 nmol/g body weight) daily for one week, followed by DSS treatment to induce colitis. **b** Changes in body weight (percentage of original body weight) over time (days) in 8-week-old male mice following DSS treatment (*n* = 4 per group). **c** Gross morphology of the large intestine. **d** Colon length on day 14 after the DSS treatment (*n* = 4 per group). **e** Hematoxylin/eosin (H&E) staining of colons and **f** pathological scores on day 14 after DSS treatment (*n* = 4 per group). Scale bars: 100 μm. **g** Measurements of 4-kDa FITC-dextran in serum on day 14 after the DSS treatment in control and *Slc39a8*-IEC KO mice (*n* = 4 per group). qPCR quantification of (**h**) tight junction and (**i**) proinflammatory cytokines and chemokines in colon mucosa on day 13 after DSS treatment in control and *Slc39a8*-IEC KO mice (*n* = 4 per group). **j** Sphingolipid metabolism. **k** Heat map showing significantly altered specific lipids in the intestines of control and *Slc39a8*-IEC KO mice with and without treatment with the ACER1 inhibitor D-e-MAPP. Shades of red and green represent upregulated and downregulated lipids, respectively (see color key). Lipid classes include SM sphingomyelins, HexCer hexosylceramides, SPB sphinganine, PE-Cer phosphatidylethanolamine ceramides, and CE: ceramides. *n* = 3 per group. **l** SMS1 activity in control and *Slc39a8*-IEC KO intestines (*n* = 6 per group) and intestinal organoid monolayer cells derived from control and *Slc39a8*-IEC KO mice (*n* = 5 biologically independent samples). **m** Average levels of three sphingomyelin species (SM 40:0;3O, SM 30:0;2O, and SM 43:1;2O). Data are presented as individual values and represent the mean ± SEM. The *p*-values were determined by two-way ANOVA with Bonferroni's multiple comparisons test for **b**, one-way ANOVA with Bonferroni's multiple comparisons test for **d**, **f**–**i**, and unpaired two-tailed Student's *t* test for **l**. Source data are provided as a Source Data file.

The phenotypic rescues by D-e-MAPP (Figs. 7–9 and Supplementary Fig. 6) suggest that the upregulation of Acer1 and associated lipidome alterations contribute to the impaired gut function in *Slc39a8*-IEC KO mice. However, the rescue effect is unlikely to be mediated by ceramides; instead, the restoration of other lipid species, such as sphingomyelin, is the candidate mediator for the observed impact of D-e-MAPP on gut physiology.

## Mn deficiency is sufficient to promote upregulation of Acer1 expression in the intestine, and treatment with an Acer1 inhibitor alleviates colitis in Mn-deficient mice

We also tested the protective effects of the Acer1 inhibitor on DSS-induced colitis in WT mice. Male C57BL/6J mice were fed either with a Mn-deficient (Mn-D) or Mn-adequate (Mn-A) diet for 14 days (Fig. 9a). The qPCR analysis revealed upregulation of *Acer1* expression in the ileal and colonic mucosa from Mn-D mice compared to Mn-A mice (Fig. 9b), indicating that Mn deficiency is sufficient to promote upregulation of *Acer1* expression in the intestine. Mice were then treated with D-e-MAPP via oral gavage (10 nmol/g at 24 h intervals) during DSS treatment, followed by placement on regular drinking water (Fig. 9a). The Mn-D mice displayed profound and sustained weight loss when compared to the Mn-A mice (Fig. 9c). Notably, D-e-MAPP treatment protected the Mn-D mice from this DSS-induced body weight loss (Fig. 9c), as well as against lower survival rate (Fig. 9d) and colon shortening (Fig. 9e, f), suggesting that the ACER1 inhibitor mitigated intestinal injury in Mn-D mice. Furthermore, the Mn-D mice treated with D-e-MAPP revealed marked amelioration of tissue inflammation and injury and significant reductions in their histological scores (Fig. 9g, h). Collectively, these results revealed Mn deficiency-induced ACER1 as a potential therapeutic target for IBDs with either genetic or environmental etiology.

## Discussion

Our studies stem from the observational studies that common genetic variants at the *SLC39A8* locus are significantly associated with various physiological traits and diseases, including whole-blood Mn levels[14] and IBD[15,16]. We used an IEC-specific *Slc39a8* KO mouse model and intestinal organoid monolayer culture to gain insight into the roles of intestinal Slc39a8 in Mn homeostasis and how Slc39a8 contributes to the pathogenesis of IBD. Our studies revealed essential roles for intestinal Slc39a8 in controlling intestinal Mn absorption and epithelial integrity. Deletion of *Slc39a8* in IEC impairs intestinal absorption of Mn, which in turn reduces the Mn content in other organs and tissues. Slc39a8 is localized to the apical surface of the ileum and colon and functions to import Mn into enterocytes. Moreover, *Slc39a8* deficiency in IECs exacerbates colitis after intestinal epithelial injury. We identified increased levels of ACER1, a lipid-generating enzyme that affects intestinal permeability, in the intestines of *Slc39a8*-IEC KO mice. We further demonstrated that treatment with an ACER1 inhibitor significantly mitigated colitis in both preventive and therapeutic settings by enhancing the expression of tight junction proteins and inhibiting inflammation. Our results demonstrate that intestinal epithelial SLC39A8 controls intestinal Mn absorption and epithelial integrity, thereby providing potential insight into therapeutic options for IBD patients with *SLC39A8* deficiency.

SLC39A8 was identified as a zinc transporter[17] and thereafter as an iron transporter[18]; therefore, the early studies focused on the characterization of zinc and iron status using cell lines overexpressing SLC39A8. In 2016, *SLC39A8* loss-of-function mutations were identified in human patients, and these patients were found to have abnormally low whole-blood Mn levels but normal levels of other metals[22,23]. Furthermore, human carriers of the *SLC39A8* A391T mutation, which is associated with reduced SLC39A8 activity, also showed reduced whole-blood Mn levels[48,49]. Thus, human genetics studies have revealed an essential role for SLC39A8 in Mn homeostasis. The in vivo roles of SLC39A8 were largely unknown; however, a study of hepatocyte-specific *Slc39a8* KO mice showed that SLC39A8 is essential for hepatic and whole-body Mn homeostasis without affecting other metal levels[25]. Two research groups independently reported alterations in Mn levels, but not zinc and iron levels, in *Slc39a8* A391T knock-in (KI) mice, although the tissue Mn levels reported in these studies were inconsistent[31,32]. In the present study, expansion of the analysis of Mn, zinc, iron, copper, and selenium in multiple tissues (intestines, liver, lung, kidney, heart, brain, and whole blood; *n* = 10 per group in both sexes) revealed that a loss of *Slc39a8* in the intestine reduced whole-body Mn levels without affecting zinc and iron concentrations in any of the tested tissues. Our findings in IEC-specific *Slc39a8* KO mice are consistent with human data showing that patients with *SLC39A8* loss-of-function mutations and human carriers of the *SLC39A8* A391T mutation have reduced whole-body Mn levels. Collectively, these studies suggest that Mn is a major physiological substrate for SLC39A8 and that SLC39A8 is required for Mn homeostasis.

One of the key findings in our present study is that SLC39A8 plays an essential role in controlling intestinal Mn absorption. Whole-body Mn homeostasis is regulated by intestinal absorption and hepatobiliary excretion[26]. The current model of intestinal Mn absorption is that enterocytes take up Mn from the diet at the apical membrane and export it into the bloodstream at the basolateral surface. The iron transporter SLC11A2, also known as divalent metal transporter 1 (DMT1), was considered to act as an intestinal Mn importer and to absorb Mn into duodenal enterocytes[50]; however, IEC-specific DMT1 KO mice showed that DMT1 is dispensable for Mn uptake in the intestine[28]. SLC39A14, a close family member of SLC39A8, localizes to the basolateral membrane of enterocytes, where it takes up Mn from the blood into the enterocytes to promote Mn elimination[51,52]. SLC30A10, a Mn exporter, localizes to the duodenal enterocyte apical membrane, where it exports Mn into the intestinal lumen for subsequent Mn elimination via feces[53,54]. These studies suggest that, in intestinal enterocytes,

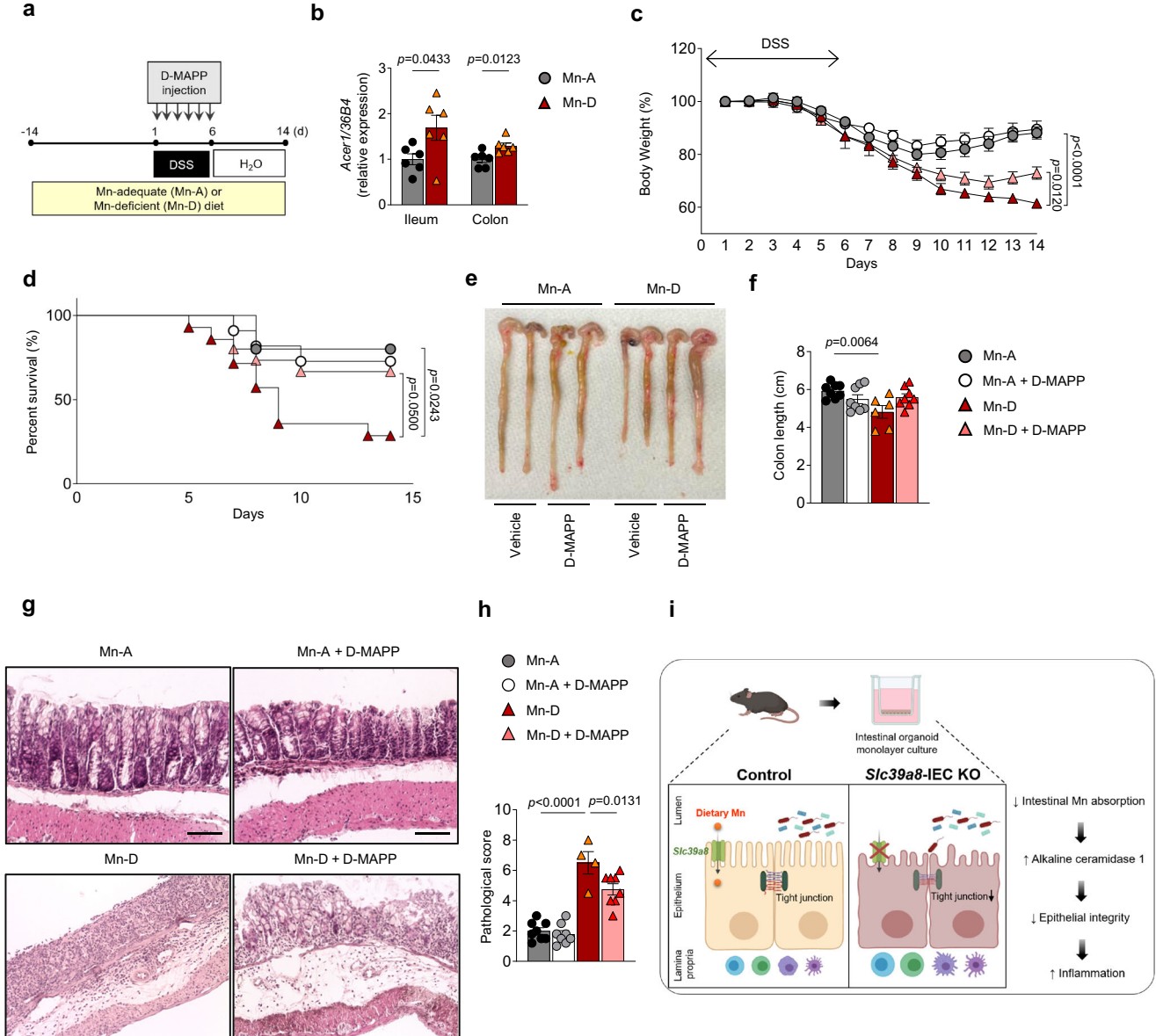

**Fig. 9 | Mn deficiency upregulates Acer1 expression and treatment with ACER1 inhibitor D-e-MAPP confers protection against colitis in Mn-deficient mice.**
**a** Schematic of the dietary Mn restriction experiment and treatment with the ACER1 inhibitor D-e-MAPP in the DSS-induced mouse model of colitis. Mice were fed either Mn-deficient (Mn-D) or Mn-adequate (Mn-A) diets for 14 days. Mice were then treated with or without D-e-MAPP (10 nmol/g body weight) daily at 24 h intervals during the DSS treatment (inflammatory phase). The mice were then provided with regular drinking water (recovery phase). **b** ACER1 expression in the ileal and colonic mucosa in Mn-A and Mn-D mice (n = 6 per group). **c** Changes in body weight (percentage of original body weight) over time (days) in Mn-A and Mn-D mice following DSS treatment (n = 15 per group). **d** Survival percentage (%) over time. **e** Gross morphology of the large intestine. **f** Colon length on day 14 after DSS treatment (Mn-A, n = 8; Mn-A + D-MAPP, n = 8; Mn-D, n = 6; Mn-D + D-MAPP, n = 8

per group). **g** Hematoxylin/eosin (H&E) staining of colons and **h** pathological scores on day 14 after DSS treatment (Mn-A, n = 8; Mn-A + D-MAPP, n = 8; Mn-D, n = 4; Mn-D + D-MAPP, n = 8 per group). Scale bars: 100 μm. **i** A model of the proposed mechanism by which Slc39a8 deficiency contributes to the pathophysiology of IBD. The proposed scheme shows that Slc39a8 deficiency impairs intestinal Mn absorption and disrupts the epithelial barrier by the upregulation of Acer1, thereby leading to inflammation. Data are presented as individual values and represent the mean ± SEM. The p-values were determined by unpaired two-tailed Student's t test for **b**, two-way ANOVA with Bonferroni's multiple comparisons test for **c**, Log-rank test for **d**, and one-way ANOVA with Bonferroni's multiple comparisons test for **f** and **h**. Source data are provided as a Source Data file. Figure 9i was created with BioRender.com released under a Creative Commons Attribution-NonCommercial-NoDerivs 4.0 International license.

SLC39A14 at the basolateral membrane and SLC30A10 at the apical membrane limit Mn absorption and enhance intestinal Mn excretion[52–54]. Our study shows that Slc39a8 localizes to the apical membrane of enterocytes, where it mediates intestinal Mn absorption.

Our data suggest that Slc39a8 mediates Mn uptake from the diet at the apical membrane, as markedly less [54]Mn was accumulated in the ileum from Slc39a8-IEC KO mice than from the intestines of controls after oral-intragastric gavage of [54]Mn. We also showed that Slc39a8-

deficient intestinal organoid monolayer cultures exhibited reduced [54]Mn uptake at the apical membrane, but not at the basolateral membrane, thereby demonstrating that SLC39A8 mediates Mn uptake at the apical membrane of enterocytes. Consistent with our findings, single-cell transcriptome analysis in human intestines has revealed high expression of SLC39A8 in both the small and large intestines, implicating the importance of these segments for metal ion absorption[55]. Thus, our data support a model in which SLC39A8 localizes to the

apical surface of the enterocytes, where it mediates the uptake of Mn from the intestinal lumen into enterocytes for subsequent intestinal Mn absorption into the bloodstream.

Two recent studies have independently reported altered Mn homeostasis and exacerbated DSS-induced colitis in *Slc39a8* A391T knockin (KI) mice[31,32]. The study by Sunuwar et al. reported that *Slc39a8* A391T KI mice showed reduced Mn levels in whole blood, liver, and kidney, but no alterations in the Mn levels in the distal small intestine, colon, lung, heart, and brain[31]. Nakata et al. reported reduced Mn levels in the whole blood, liver, and colon of *Slc39a8* A391T KI mice, but they did not examine Mn levels in other tissues[32]. The colon Mn data reported in these two studies were inconsistent. Both studies measured the steady-state levels of metals by ICP-MS but did not directly measure intestinal Mn absorption using radio-tracer studies. In addition, both studies investigated whole-body mutant mice, which developed a minimal phenotype, but neither study directly explored the cell-type-specific mechanism linking *Slc39a8* deficiency to impaired intestinal barrier function. Our present study demonstrates the roles of intestinal SLC39A8 in the Mn supply for all peripheral tissues examined. Thus, our study provides insights into how Slc39a8 contributes to Mn absorption and protect animals from colitis.

Our RNA-seq analysis resulted in the identification of four differentially expressed genes in the *Slc39a8*-IEC KO intestines (Fig. 6a and Supplementary Table 1). Among these four genes, *Slc39a8* was the only downregulated gene in the *Slc39a8*-IEC KO intestines (Fig. 6a and Supplementary Table 1), validating the RNA-seq approach. *Acer1* encodes alkaline ceramidase 1 (ACER1), which catalyzes the hydrolysis of ceramides to generate sphingosine[56,57]. Acer1 catalyzes the hydrolysis of ceramides with unsaturated long acyl chains (C18:1 and C20:1) or a very long saturated (C24:0) or unsaturated (C24:1) acyl chain[57]. Considering the significant association between sphingolipid metabolism and IBD[41], we further investigated ACER1 in our study. In addition to Acer1, two other members of the alkaline ceramidase family, Acer2 and Acer3, are present in both mice and humans[57,58]. D-*e*-MAPP can inhibit ACER2 and ACER3[59,60]; however, since we did not find any expression changes in *Acer2* and *Acer3* in *Slc39a8*-IEC KO mice (Supplementary Fig. 8a–e), the rescue effect observed with D-*e*-MAPP was likely due to ACER1 inhibition. Nevertheless, the lack of expression changes still does not rule out a potential involvement of Acer2 and Acer3. A previous study has demonstrated that ACER3 plays a crucial role in mediating the immune response in innate immune cells and colonic epithelial cells[61]. Acer3 deficiency has been shown to increase the local and systemic production of proinflammatory cytokines and exacerbated colitis in the DSS-induced murine colitis model[61]. These observations indicate that Acer activities are tightly regulated to ensure normal gut functions.

In addition to ACER1's role in sphingosine production, sphingosine serves as the substrate for sphingosine kinases (SphKs), which have been extensively implicated in IBD[41,42]. Thus, we tested their expression levels in our model. We found a significant upregulation of *Sphk1* mRNA expression in the ileum and colon of *Slc39a8*-IEC KO mice, whereas *Sphk2* expression levels remain unchanged (Supplementary Fig. 8f, g), suggesting the involvement of the SphK1/S1P pathway. However, our lipidomics data indicate no changes in sphingosine, likely due to compensatory mechanisms. Future studies are needed to explore the intricate regulation of sphingosine metabolism in response to *Slc39a8* deficiency.

One important implication of our findings is that the use of an ACER1 inhibitor might be a better therapeutic avenue than dietary Mn supplementation because our data suggest that *Slc39a8* deficiency impairs intestinal Mn absorption from diet. In addition to the findings for *Slc39a8*-IEC KO intestines, our data demonstrate that dietary Mn deficiency upregulates *Acer1* expression in the intestine and that treatment with an Acer1 inhibitor mitigates colitis in Mn-

deficient mice (Fig. 9). Epidemiological studies have shown a positive relationship between Mn deficiency and the risk of IBD. A recent epidemiological survey showed that pediatric patients with newly diagnosed Crohn's disease and ulcerative colitis have significantly lower Mn levels in their hair samples[6]. Another human study reported significantly lower blood Mn levels in patients with active Crohn's disease and ulcerative colitis compared to those in remission[62]. These data suggest that IBD patients have a risk of Mn deficiency, and may consequently upregulate Acer1 expression; this possibility can be tested in the future. Nevertheless, interventions aimed at restoring ACER1 levels may be beneficial in improving human IBD conditions. Our study has also revealed an involvement of sphingolipid metabolism and its therapeutic potential in IBD. Further studies are needed to examine whether aberrant sphingolipid metabolism is a general mechanism of IBD or whether it defines a subset of IBD conditions.

## Methods

### Animals

All animal experiments were reviewed and approved by the Institutional Animal Care and Use Committee of the University of Michigan (PRO00008963). For IEC-specific disruption of *Slc39a8*, mice floxed for *Slc39a8* (*Slc39a8^{fl/fl}*) on a C57BL/6J background (European Mouse Mutant Archive, EM: 05285) were crossed with *Villin-Cre* (*Vil-Cre*) mice on a C57BL/6J background (Jackson Lab, stock number 004586). The *Slc39a8-flox* allele was detected by PCR using the following primers: Forward: 5'-AAGGCGCATAACGATACCAC-3' and Reverse: 5'-CCGCCTACTGCGACTATAGAGA-3'. The *Villin-Cre* transgenic allele was detected using the following primers: Forward: 5'-TTCTCCTCTAGGCTCGTCCA-3' and Reverse: 5'-CATGTCCAT-CAGGTTCTTGC-3'. The *Slc39a8*-IEC KO (*Slc39a8^{fl/fl}:Villin-Cre*) and *Slc39a8*-IEC Het (*Slc39a8^{fl/+}:Villin-Cre*) mice were compared to controls, which included a mixture of three groups: *Slc39a8*-flox (*Slc39a8^{fl/fl}* and *Slc39a8^{fl/+}*), *Slc39a8*-WT (*Slc39a8^{+/+}*), and *Vil-Cre* (*Slc39a8^{+/+}:Villin-Cre*). We compared key experimental results, such as metal concentrations and experimentally induced colitis models, among the four genotypes of controls. We found no significant differences in Mn concentrations in the ileum and colon among the four genotypes of controls at 6–8 weeks of age (Supplementary Fig. 9a). We also observed that the four genotypes of controls did not display significant differences in DSS sensitivity at 6–8 weeks of age (Supplementary Fig. 9b). These data indicate that the flox sequence and Cre transgene did not impact Mn homeostasis or DSS sensitivity. The mice were housed in a pathogen-free animal facility at 22 °C with 40–60% humidity on a 12-h light/dark cycle, and provided the standard rodent diet of our institution (PicoLab Laboratory Rodent Diet 5LOD, LabDiet; 70 ppm Mn) and water *ad libitum*.

For the Mn-restriction diet experiment, WT C57BL/6 mice aged 4 weeks were purchased from Jackson Laboratory and maintained on a metal-basal diet containing 35 mg Mn/kg (TD120518, Harlan Teklad, Indianapolis, IN, USA)[13,27,63]. The trace element levels in the diet were as recommended by the American Institute of Nutrition[64]. For dietary Mn alterations, the mice were fed either Mn-deficient or Mn-adequate diets (<0.01 and 35 ppm Mn, respectively; Harlan Teklad)[13]. For colitis induction, mice were provided with water containing 3% (w/v) DSS for 6 days (inflammatory phase). The mice were then placed on regular drinking water for 8 days (recovery phase). All mice in the same experiment were from an age-matched, co-housed cohort. This study was performed in strict accordance with the recommendations in the Guide for the Care and Use of Laboratory Animals of the National Institutes of Health (Bethesda, MD, USA).

### Materials

General laboratory chemicals and reagent solutions were purchased from Sigma-Aldrich. The ACER1 inhibitor D-*erythro*-MAPP was

purchased from Enzo Life Sciences. Specific doses of D-erythro-MAPP were chosen based on a previous study[65]. ELISA kits for AST and ALT were purchased from Sigma Aldrich.

## Metal measurements

Tissue samples were analyzed for metals by inductively coupled plasma mass spectrometry (ICP-MS)[13,20,24]. Briefly, tissue samples taken from mice were digested with 2 mL/g total wet weight nitric acid (BDH ARISTAR® ULTRA) for 24 h and then digested with 1 mL/g total wet weight hydrogen peroxide (BDH ARISTAR® ULTRA) for 24 h at room temperature. Specimens were preserved at 4 °C until quantification of metals. Ultrapure water was used for final sample dilution.

## Absorption of Mn in vivo

To assess radiotracer Mn absorption, mice were dosed with 0.1 μCi $^{54}$Mn (PerkinElmer) per gram body weight via oral-intragastric gavage or tail vein injection. Blood was collected at 15 min, and mice were killed at 4 h, after which tissue $^{54}$Mn-associated radioactivity (counts per minute) was determined using a γ-counter[27,63].

## Intestinal organoid monolayer cultures

Intestinal organoid monolayer cultures were created and maintained as previously described[29,66]. Briefly, mouse ileum or colon was dissected and flushed with phosphate buffered saline (PBS) and transferred to chelation buffer (2 mM EDTA, PBS) for 30 min. The ileum or colon was shaken to remove crypt cells and then incubated in PBS with 43 mM sucrose and 55 mM sorbitol and filtered through a 70-μm filter. Isolated intestinal crypts from mice were seeded and maintained in L-WRN conditioned complete media supplemented with 50 ng/mL recombinant human EGF (R&D Systems) and an antibiotic–antimycotic (Corning).

## $^{54}$Mn uptake study

Intestinal organoid monolayer cultures were grown on Transwell inserts, and $^{54}$Mn uptake assay was determined as previously described[20,24]. During the $^{54}$Mn uptake experiment, the extracellular pH was maintained at pH 7.4 in the apical and basolateral compartments to mimic the physiological situation (i.e., the neutral intraluminal pH of the terminal ileum and colon and the normal blood pH)[67]. Apical (basolateral) metal uptake experiments were initiated by adding 1 μM $^{54}$Mn to the apical (basolateral) chamber, and the cells were incubated at 37 °C at the indicated time points. After uptake, cells were washed 3 times with PBS containing 1 mM EDTA to remove any unbound $^{54}$Mn and directly harvested for intracellular radioactivity. Radioactivity was determined with a γ-counter and was normalized to the cell protein measured in lysates using the Bradford assay.

## Immunofluorescence

Immunofluorescence staining was performed on frozen tissues or intestinal organoid monolayer cultures[20,24]. Primary antibodies were incubated at room temperature for 1 h, followed by fluorescent secondary antibodies at room temperature for 20 min. Nuclei were detected with 4′,6-diamidino-2-phenylindole (DAPI). The following primary antibodies were used: SLC39A8 (Proteintech, 20459-1-AP, 1:50), ZO-1 (Santa Cruz, sc-33725, 1:50), ZO-2 (Santa Cruz; sc-515115, 1:50), Claudin-2 antibody (Thermo Fisher, # 51-6100, 1:100), Claudin-3 antibody (Thermo Fisher, # 34-1700, 1:100), Claudin-5 antibody (Thermo Fisher, # 35-2500, 1:50), Claudin-7 antibody (Thermo Fisher, # 34-9100, 1:50), and Occludin antibody (Santa Cruz, sc-133256, 1:50). The following secondary antibodies were used: Donkey anti-rat IgG H&L (Alexa Fluor 488) (Thermo Fisher, #A48269, 1:500), Donkey anti-mouse IgG H&L (Alexa Fluor 488) (Thermo Fisher, # A32766, 1:1000), Donkey anti-rabbit IgG H&L (Alexa Fluor 488) (Thermo Fisher, # A32790, 1:000), Goat anti-mouse IgG H&L (Alexa 488 Fisher A11001, 1:1500), and Goat anti-rabbit IgG H&L (Alexa 568 Fisher A11036, 1:1500), Confocal microscopy was performed using a Nikon Eclipse A-1 confocal microscope (Nikon Instruments). Image analysis was performed using ImageJ (NIH) software.

## Chemically induced colitis models

For the acute DSS model, mice were treated with 3% DSS (Fisher Scientific) dissolved in the drinking water to induce colitis (inflammatory phase) and then placed on regular drinking water (recovery phase). The mice were sacrificed on Day 13 or 14 after DSS treatment. For the acute TNBS model, mice were intrarectally injected with 100 mg/kg TNBS (MilliporeSigma) in 50% ethanol, with 50% ethanol treatment used as a control. The mice were sacrificed 7 days after TNBS treatment. For the chronic DSS model, mice received two cycles of 3% DSS treatment, each cycle consisting of 8 days of inflammatory phase followed by 5 days of recovery phase. The mice were sacrificed on Day 26 after DSS treatment. Body weights, stool consistency, and GI bleeding were monitored daily. Clinical scores and colon damage scores were estimated as detailed previously[39].

## Histology and immunohistochemical staining

Tissues were fixed in 4% paraformaldehyde at 4 °C overnight. Hematoxylin/eosin (H&E) analysis was performed in paraffin-embedded tissue sections (5 μm). The histological scoring was performed by a pathologist who blindly assessed the degree of surface epithelial loss, crypt destruction, and inflammatory cell infiltration into the mucosa, resulting in a score from 0 (no disease activity) to 12 (most severe disease activity).

## Intestinal permeability assay

Intestinal permeability assay was performed as previously described[13]. Briefly, fluorescein isothiocyanate (FITC)-dextran (MW 4000; FD4, Sigma-Aldrich) was administered by oral gavage to examine the intestinal barrier function. Serum was collected 4 h later, and the fluorescence intensity of each sample (excitation, 485 nm; emission, 525 nm) was measured using a BioTek Synergy microplate reader (BioTek Instruments).

## Transepithelial electrical resistance (TEER)

IECs were grown on permeable supports, and monolayers were monitored for electrical resistance using an epithelial volt-ohmmeter (EVOM/EndOhm; World Precision Instruments).

## RNA isolation and sequencing

Total RNA was isolated from ileal or colonic mucosa from each mouse using the RNeasy mini kit (Qiagen, Valencia, CA). Three mice (n = 3 biological replicates) were used per experimental group. The sample size was determined based on previously described power calculations to optimize the detection of differentially expressed genes[68]. RNA concentrations were measured with an Epoch Microplate Spectrophotometer (BioTek Instruments). RNA integrity was assessed with an Agilent Bioanalyzer 2100 using a Nano 6000 assay kit (Agilent Technologies). An RNA integrity number (RIN) > 7.2 was considered the minimum requirement for library preparation. RNA was reverse transcribed into cDNA using oligo-dT, and cDNA libraries were generated with a NEBNext Ultra II RNA Library Prep Kit (NEB #E7775). An insert size of 250–300 bp was used for cDNA library preparation. Libraries were sequenced on the Illumina NovaSeq 6000 platform with a 150-bp paired-end mode. Reference genome and annotation files were downloaded from Ensemble, and RNA-seq data were aligned to the reference genome using the Spliced Transcripts Alignment to a Reference (STAR) software[69]. The DESeq2 package was used for differential expression analysis.

## qPCR

Purified RNA was reverse-transcribed with SuperScript III First-Strand Synthesis System (Invitrogen, Thermo Fisher Scientific)[70]. The qPCR was performed using the Power SYBRGreen PCR Master Mix (Applied Biosystems). The mRNA was normalized using 36B4. The primers used for qPCR are listed in Supplementary Table 2, and were all purchased from Integrated Genomics Technologies.

## Immunoblot

Protein samples were loaded on SDS-polyacrylamide gels, separated by electrophoresis, and then transferred to a nitrocellulose membrane (Bio-Rad, Hercules, CA, USA; Cat. No. 1620115). The membrane was immunoblotted with primary antibody ZO-1 (Santa Cruz, sc-33725, 1:500), ZO-2 (Santa Cruz; sc-515115, 1:500), Claudin-2 (Thermo Fisher, # 51-6100, 1:1000), Claudin-3 (Thermo Fisher, # 34-1700, 1:1000), Claudin-5 (Thermo Fisher, # 35-2500, 1:1000), Claudin-7 (Thermo Fisher, # 34-9100, 1:1000), Occludin (Santa Cruz, sc-133256, 1:1000), Cytokeratin 8/18 (Abcam, ab53280, 1:50000), ASAH3 antibody (Thermo Fisher, # PA5-75603, 1:500), and Actin (Proteintech, 66009-1-lg, 1:3000). The blots were visualized with infrared secondary antibodies, IRDye 680RD Goat anti-rat IgG Secondary antibody (Liborbio, #926-69076, 1:20000), IRDye 800CW Donkey anti-mouse IgG Secondary antibody (Liborbio, #925-32212, 1:20000), or IRDye 680LT Donkey anti-rabbit IgG Secondary antibody (Liborbio, #926-69021, 1:15000), using a LI-COR Odyssey fluorescent western blotting system (LI-COR Biosciences, Lincoln, NE, USA). Protein expression was quantified by densitometry (Image Studio Lite; LI-COR).

## Lipidomics

Details of sample preparation and identification for untargeted lipidomic profiling have been previously reported[71]. Lipids were extracted using a modified Bligh-Dyer method[72]. Extraction was carried out using water:methanol:dichloromethane (2:2:2, v/v) at room temperature after spiking with internal standard lipids. The organic layer was collected and dried completely under nitrogen and the contained lipids were further analyzed by liquid chromatography-mass spectrometry (MS)-based lipidomics. The dried lipid extracts were injected onto a 1.8-μm particle 50 ×2.1 mm id Waters Corporation Acquity HSS T3 column, which was heated to 55 °C. A binary gradient system consisting of acetonitrile and water with 10 mM ammonium acetate (40:60, v:v) was used as eluent A. Eluent B consisted of water, acetonitrile, and isopropanol, both containing 10 mM ammonium acetate (5:10:85, v/v). The lipid extracts were reconstituted in buffer B and injected for MS. The MS analysis alternated between MS and data-dependent MS2 scans using dynamic exclusion in both positive and negative polarity. As quality controls (QC) to monitor the profiling process, pools of plasma and test plasma (a small aliquot from all test samples) were extracted and analyzed in tandem with experimental samples. These controls were incorporated multiple times into the randomization scheme to allow constant monitoring of the sample preparation and analytical variability.

Lipids were identified using the LIPIDBLAST library[73] by matching the product ion MS/MS data. To facilitate accurate lipid identification, a more stringent mass error rate of 0.001 m/z for the positive mode and 0.005 m/z for the negative mode was determined based on the mass accuracy of internal standards to reduce the rate of false positives and identify likely correct candidates. The sphingolipid species considered for identification were limited to features with (%RSD < 20) in pooled samples. More details regarding the spectral matching procedure used for compound identification against the LipidBlast library are outlined in the original report[74].

A manual review of spectral matches was performed in all potentially uncertain cases, including the sphingolipid species. Examples of spectral matches are included in Supplementary

Fig. 10a, b. The product ion scan of [M+Na]+ species of sphingomyelin (SM 34:1, 2O) yields three fragments corresponding to neutral loss (NL) of C3H9N (m/z 666.477, C36H70NO6PNa+), NL of C5H14NO4P (m/z 542.486, C34H65NO2Na+), NL of C5H16NO5P (m/z 502.497, C34H64NO+), respectively. A product ion scan of m/z 300.2860, which corresponds to the [M + H]+ ion for sphingosine (SPB) 18:1;2O molecular species, was searched against the LipidBlast In-Silico library, revealing an exact match for the expected compound with high library scores (Supplementary Fig. 10a). MS/MS spectra of ceramides in the positive mode were dominated by fragments from the sphingoid base (Supplementary Fig. 10b). Collisional activation of ceramides in positive ion mode showed the characteristic product ions of m/z 264.2684 [C18H34N]+ and 282.279 that were attributed to the loss of N-linked fatty acid and one and two water molecules, respectively. The positive mode MSMS product ion spectrum of HexCer 18:0;3O/16:0;(2OH), which is illustrated in Supplementary Fig. 10b. This compound shows an NL of C6H10O5, the hexose ring, results m/z 572.5232 (Supplementary Fig. 10b), as well as other peaks that are m/z 554.5143 and m/z 264.2686 correspond to NL of C6H10O5 and H2O and SPB 18:0;O3 −3H2O respectively. Chromatographic retention time (RT) was utilized to validate identifications. The scatter plots (Supplementary Figs. 10c, d) demonstrate a linear relationship between sphingolipid RT, alkyl chain length, and desaturation, aligning with expected elution patterns. No significant deviations were observed, supporting the accuracy of our identifications.

MS data files were processed using MultiQuant 1.1.0.26 (Applied Biosystems/MDS Analytical Technologies). Identified lipids were quantified by normalizing against their respective internal standard. QC samples were used to monitor the overall quality of the lipid extraction and MS analyses and were mainly used to remove technical outliers and lipid species that were detected below the lipid class-based lower limit of quantification.

## Statistics and reproducibility

Data are presented as individual values and represent the mean ± SEM. The exact sample size and the statistical tests are described in figure legends, exact $p$ values are provided in the figure legends and Source Data. Test used include one-way ANOVA, two-way ANOVA, or two-tailed Student's $t$ test by GraphPad Prism software 8.0 (San Diego, CA, USA). Outliers were identified using the GraphPad ROUT (robust regression and outlier removal) method (Q = 1%).

## Reporting summary

Further information on research design is available in the Nature Portfolio Reporting Summary linked to this article.

## Data availability

All data generated or analyzed during this study are included in this published article (and its supplementary information files). All the datasets in this study are existing published and are available via the NCBI website, including Gene Expression Omnibus (GSE) accession number: GSE192695. Lipidomics data have been deposited at https://github.com/SeoResearchLab/IECKO2023. Source data are provided with this paper. Any additional information is available upon request to the corresponding author. Source data are provided with this paper.

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

## Acknowledgements

This work was supported by grants from the National Institutes of Health (R01DK123022 and R21NS112974) to Y.A.S.; (R01NS116008 and R01NS089896) to S.I. and from the University of Michigan Center for Gastrointestinal Research (DK034933) to Y.A.S. We acknowledge Theresa Keeley in the lab of Dr. Linda Samuelson for assistance with the immunofluorescence studies.

## Author contributions

Y.A.S. conceived and designed the study. E.K.C., T.M.R., T.S., J.H.P., S.I., and Y.A.S. acquired the data. E.K.C., T.M.R., TS, J.H.P., C.K.M., V.G.H., N.K., T.S., T.M.R., L.C.S., A.N., S.I., and Y.A.S. developed the methodologies. E.K.C., T.M.R., T.S., J.H.P., T.M.R., S.I., and YAS performed experiments. E.K.C., T.M.R., T.S., J.H.P., L.A., N.K., L.C.S., A.N., S.I., and Y.A.S. analyzed and interpreted the data. YAS wrote the manuscript. Y.A.S. supervised the study. Y.A.S. was responsible for funding acquisition. Figures 3a, c, e, and 9i were created with BioRender.com released under a Creative Commons Attribution-NonCommercial-NoDerivs 4.0 International license.

## Competing interests

The authors declare no competing interests.
