## [Peer Review File · Nature Communications]

The essential role of SLC39A8 in intestinal manganese absorption links alkaline ceramidase 1 to inflammatory bowel diseaseREVIEWER COMMENTS

Reviewer #1 (Remarks to the Author):

This manuscript by Choi et al. aimed to investigate SLC39A8 in intestinal manganese absorption links alkaline ceramidase 1 to inflammatory bowel disease (IBD). SLC39A8 is associated with physiological traits and diseases, including blood manganese (Mn) level and IBD. In the current study, they generated Slc39a8 intestinal epithelial cell-specific-knockout (Slc39a8-IEC KO) mice, which displayed markedly decreased Mn levels in most organs and whole blood. Radiotracer studies revealed that Slc39a8-IEC 28 KO mice had impaired intestinal absorption of dietary Mn. SLC39A8 localized to the apical membrane and mediated Mn uptake in intestinal organoid monolayer cultures. Unbiased transcriptomic analysis identified alkaline ceramidase 1 (ACER1), a key enzyme in the biogenesis of sphingosine, as a potential therapeutic target for SLC39A8-associated IBDs. Importantly, treatment with an ACER1 inhibitor attenuated colitis in Slc39a8-IEC KO mice by remedying barrier dysfunction. These results highlight the essential roles of SLC39A8 in intestinal Mn absorption and epithelial integrity and offer the first therapeutic target for IBD associated with impaired Mn homeostasis.

The authors presented comprehensive epithelial functions associated with the risk of inflamed intestine. This study is interesting and provides new insights into the involvements of the Mn homeostasis and epithelial integrity in the development and therapy of IBD.

There are some comments for further improving the current manuscript.

1. Figure 1. Only female mice were used in Fig. 1A and only male mice were used in Fig. 1B. Please explain any gender differences observed in the mouse model?
2. For Figure 4F, H&E data should include mice with and without DSS. The Fig. G-J should provide the data of Control and KO mice with or without DSS treatment.
3. In Figure 4, the mRNA data on selective genes for TJ proteins are not sufficient to show the changes of TJs. The authors need provide western blots and immunostaining data to show the protein levels and locations of TJ proteins. Claudin-2 is a "leaky protein" known increased in human IBD and colitis models. Why the mRNA level of Claudin2 decreased more in the DSS treated KO mice, compared to the controls.
4. The Fig. 5C-F, the authors need provide the data of Control and KO mice with or without DSS treatment.
5. Fig. 6. The protein expression data of ACER1 is needed for validation of ACER1 expression.
6. Fig. 7. The authors need provide data of protein levels of TJ proteins, e.g. ZO-1 and Claudins, in addition to the PCR data.
7. Will ACER1 inhibitor rescue the IL-10 KO mice?
8. More discussion should be added on human IBD if the authors believe that Mn deficiency upregulates Acer1 expression in the DSS-colitis model.
9. A working model of ACER1, Mn deficiency, and barrier in the development of chronic intestinal inflammation will help the readers.

Reviewer #2 (Remarks to the Author):

Choi et al. have submitted a solid manuscript that interrogates the role of the manganese transporter Slc39a8 in intestinal manganese absorption and inflammatory bowel disease using cell culture and mouse models. The work is highly original and will have a prominent impact on the field of manganese biology and intestinal disease. We do have several recommendations on how the manuscript can be improved. These are described here:

-We highly recommended separating males and females for each panel instead of pooling them together. Even though no sex-specific differences are observed for Mn levels in Fig. 1, this doesn't necessarily mean such differences do not occur for other phenotypes. For instance, in Fig. S4, the authors show that female mice have lower sensitivity to effects of DSS. Additionally, the authors point out in the discussion that "inconsistent Mn concentrations among the two studies (#39,40) [of Slc39a8 A391T KI mice] complicate any interpretations of how SLC39A8 deficiency alters Mn homeostasis and exacerbates DSS-induced colitis"—could sex-specific differences contribute to the inconsistencies in these studies?

-This study used a mixture of three genotypes as a control (Slc39a8-flox (Slc39a8^{fl/fl} and Slc39a8^{fl/+}), Slc39a8-WT (Slc39a8^{+/+}), and Vil-Cre (Slc39a8^{+/+}:Villin-Cre)). Authors should not use four different genotypes as controls. This is a key issue that needs to be addressed.

-For the 54Mn gavage experiments, the authors collected blood 15 minutes after gavage and tissues four hours after gavage. Four hours is more than enough time for 54Mn to have undergone biliary excretion and enterohepatic circulation. Can the authors comment on why tissues weren't harvested at the same time as blood and how this discrepancy would influence their results?

-It appears that the confocal immunofluorescence images in Fig. 3b are misaligned—ZO-1 staining crosses through nuclei!

-Were there any changes in expression of Dmt1 or other relevant metal transporters in the intestines of the Slc39a8 IEC KO mice?

-Ceramide accumulation is known to mediate inflammation (<https://www.nature.com/articles/nm1748>). ACER1 inhibition will result in ceramide accumulation according to the pathway shown in Fig. 6c, so would ACER1 inhibition be expected to exacerbate colitis? (Also, alkaline ceramidase 3 deficiency aggravates colitis and colitis-associated tumorigenesis in mice by hyper-activating the innate immune system <https://www.nature.com/articles/cddis201636>). Please comment on this.

-Did the authors measure ceramide and sphingosine levels in cell culture and mouse models with and without Acer1 inhibition? This would help to confirm the physiologic relevance of altered Acer1 expression and the efficacy of Acer1 inhibition.

-The authors state that the increased spleen mass in Slc39a8 IEC KO mice implicates splenic macrophage infiltration. Was splenic macrophage infiltration observed by histology?

-The histological images included in figures are quite small. The authors are recommended to include images of the same magnification and large enough to visualize differences between images.

-It is striking that only four genes were found to be differentially expressed in RNA seq analysis. Can the authors confirm that this is correct? Also, do the genes other than Acer1 have potential implications for the study at hand? What is known about these other genes? These issues need to be addressed.

-It appears that a legend is missing from Fig. 9B.

-The authors are recommended to include a model figure summarizing their results.

-The authors suggest that Mn deficiency reduces Cpes activity in the discussion about how Slc39a8 IEC KO mice develop misregulated Acer1 gene expression. Can the authors measure Cpes activity in their cell culture and mouse models?

Reviewer #3 (Remarks to the Author):

The authors present data demonstrating a role for SLC39A8 in manganese (Mn) absorption. Intestinal specific deletion of SLC39A8 resulted in decreased Mn in tissues and circulation and rendered mice more susceptible to intestinal injury with dextran sulfate sodium. This loss also lead to an increase in ACER1 mRNA in intestinal tissues.

The authors then demonstrate D-e-MAPP (an alkaline ceramidase inhibitor) partially protects from DSS-induced intestinal injury.

The work is well done and well controlled; however, a critical supplemental experiment is missing and some critical points are missing from the discussion.

The authors should check the mRNA levels of the other ceramidases (there are 5 total - 1 neutral, 1 acid, and 3 alkaline).

D-e-MAPP inhibits all 3 alkaline ceramidases. This is one of the critical points missing in the discussion. It is a specific alkaline ceramidase inhibitor at the doses used, but inhibits more than ACER1.

There are also studies using alkaline ceramidase 3 (ACER3) deficient mice in colitis and colitis associated cancer. This is not included in the discussion, but likely should be.

Response to reviewers

We thank the reviewers for the thoughtful comments and suggestions. As detailed in a point-by-point response below, we have addressed all of them with additional experiments, and we feel that the manuscript has been substantially strengthened as a result.

Reviewer #1

The authors presented comprehensive epithelial functions associated with the risk of inflamed intestine. This study is interesting and provides new insights into the involvements of the Mn homeostasis and epithelial integrity in the development and therapy of IBD. There are some comments for further improving the current manuscript.

We sincerely appreciate the reviewer's positive feedback and acknowledgment of the significance of our study. We value the reviewer's comments and suggestions for further improving our manuscript.

1. *Figure 1. Only female mice were used in Fig. 1A and only male mice were used in Fig. 1B. Please explain any gender differences observed in the mouse model?*

We appreciate the reviewer's comment and the opportunity to address it. We conducted additional experiments using both female and male mice to investigate potential gender differences in *Slc39a8* expression and localization in the intestines. We performed qPCR analysis of *Slc39a8* expression in male mice, in addition to the female mice analyzed in Fig. 1a. The results from male mice showed similar patterns of *Slc39a8* expression, with higher levels observed in the distal small intestine and colon (**Fig. 1a**). We also performed immunofluorescent *Slc39a8* staining in female mice, in addition to the male mice analyzed in Fig. 1b. The results from female mice revealed a similar localization of *Slc39a8* in the apical membrane of enterocytes (**Fig. 1b**). These results indicate no sex-specific differences in *Slc39a8* expression and localization in the intestines. These new data have been added to **Fig. 1a** and **Fig. 1b** and are indicated in the text and Figure legends.

2. *For Figure 4F, H&E data should include mice with and without DSS. The Fig. G-J should provide the data of Control and KO mice with or without DSS treatment.*

This is another important point. Our new analysis showed that, without DSS treatment, no significant differences were observed in H&E staining (**Fig. 4f**), total pathological score (**Fig. 4g**), 4 kDa FITC-dextran permeability (**Fig. 4h**), qPCR quantification of tight junction markers (**Fig. 4i**), or proinflammatory cytokine and chemokine expression (**Fig. 4l**) between control and *Slc39a8*-IEC KO mice. We have updated **Fig. 4f, 4g, 4h, 4i, and 4l**, and indicated these findings in the text and Figure legends.

3. *In Figure 4, the mRNA data on selective genes for TJ proteins are not sufficient to show the changes of TJs. The authors need provide western blots and immunostaining data to show the protein levels and locations of TJ proteins. Claudin-2 is a "leaky protein" known increased in human IBD and colitis models. Why the mRNA level of Claudin2 decreased more in the DSS treated KO mice, compared to the controls?*

We agree with the reviewer and have now provided immunofluorescence and immunoblot blot analysis of tight junction proteins. The expression levels and localization of the tight junction proteins

were examined by immunofluorescence. Specifically, Zo1, Zo2, Cldn2, Cldn5, and Ocln were observed at the apical side of the colonic epithelium, whereas Cldn3 and Cldn7 were found at the lateral membrane in control mice (**Fig. 4j and Supplementary Fig. 3e**). The localization and expression of these tight junction proteins were severely impaired by DSS treatment (**Fig. 4j**). Notably, the expression levels of Cldn3 and Cldn5 were significantly reduced in the *Slc39a8*-IEC KO mice after DSS treatment (**Fig. 4j**). Immunoblotting analysis confirmed the diminished levels of Cldn3 and Cldn5 (**Fig. 4k**). These data indicate a selective impact of *Slc39a8* on the transcripts and proteins related to tight junctions. These new data have been included in **Fig. 4j, 4k, and Supplementary Fig. 3e**, and the relevant information has been added to the text and Figure legends.

As the reviewer points out, Claudin 2 protein levels generally increase in inflammatory states in human tissues^{1,2}. By contrast, we observed a reduced mRNA level of Claudin 2 in DSS-treated *Slc39a8*-IEC KO mice compared to the controls (Fig. 4i). To address this question, we carefully performed additional experiments. In these new experiments, we specifically evaluated the Claudin 2 protein levels instead of mRNA expression to obtain more conclusive results.

We thought that the apparent Claudin 2 decrease was due to DSS-induced epithelial cell loss. To test this, we normalized the Claudin 2 western blot signal against the level of Keratin, an epithelial marker. This analysis indicated that the normalized Claudin 2 levels were still decreased in control mice after DSS treatment (**Fig 4k and Supplementary Fig. 3f**). The *Slc39a8*-IEC KO mice exhibited a more pronounced reduction of Claudin 2 levels following DSS treatment, although the difference did not reach statistical significance (**Supplementary Fig. 3f**). We also performed immunofluorescence labeling to visualize the localization and expression of Claudin 2. Our immunofluorescence analysis demonstrated impaired localization and reduced expression of Claudin 2 in the control mice following DSS treatment (**Fig. 4j and Supplementary Fig. 3e**), with more pronounced effects observed in *Slc39a8*-IEC KO mice, although the differences again did not reach statistical significance (**Fig. 4j and Supplementary Fig. 3e**).

While we cannot definitively explain the reasons for the reduced levels of Claudin 2 in both control and *Slc39a8*-IEC KO mice after DSS treatment, we have found supporting evidence from similar studies that demonstrate the downregulation of Claudin 2 in wild-type C57BL/6 mice following DSS treatment^{3,4}. The discrepancy could stem from species differences or variations in experimental conditions, and this merits future investigation. Overall, these additional experiments and observations highlight the complex regulation of Claudin 2 in the context of DSS-induced inflammation. We have included these additional data in **Fig. 4j, 4k, and Supplementary Fig. 3f, 3e** and referenced them in the text and Figure legends accordingly.

4. *The Fig. 5C-F, the authors need provide the data of Control and KO mice with or without DSS treatment.*

Our new analysis showed that, in the absence of DSS treatment, no significant differences were observed in colon length (**Fig. 5c**), survival percentage (**Fig. 5d**), H&E staining (**Fig. 5e**), or total pathological scores (**Fig. 5f**) between control and *Slc39a8*-IEC KO mice. We added these data to **Fig. 5c, 5d, 5e, and 5f** and indicated them in the text and Figure legends.

5. *Fig. 6. The protein expression data of ACER1 is needed for validation of ACER1 expression.*

We conducted immunoblot analysis of ACER1 in the ileum and colon of both control and *Slc39a8*-IEC KO mice, as well as in the enteroid and colonoid monolayers derived from these mice. The immunoblotting results confirmed a significant increase in Acer1 protein levels in the ileum (~1.4

fold, $P < 0.05$) and colon (~1.3 fold, $P < 0.05$) of *Slc39a8*-IEC KO mice and the enteroid monolayers (~4.5 fold, $P < 0.01$) of *Slc39a8*-IEC KO mice. Although the increases in the levels of the Acer1 protein in the colonoid monolayers (1.3 fold) of *Slc39a8*-IEC KO mice did not reach statistical significance, the smaller changes in the colon were consistent with the RNA-seq results. We have added these new data to the manuscript as **Fig. 6e and 6f** and indicated the changes in the text and figure legends.

6. *Fig. 7. The authors need provide data of protein levels of TJ proteins, e.g. ZO-1 and Claudins, in addition to the PCR data.*

In response to this comment, we conducted immunoblot analysis to assess the protein levels of tight junction proteins ZO-1, Claudin 3 (Cldn3), Claudin 5 (Cldn5), and Claudin 7 (Cldn7) in the enteroid monolayers derived from *Slc39a8*-IEC KO mice and control mice. The immunoblot analysis confirmed that D-e-MAPP treatment enhanced levels of tight junction proteins ZO-1, Cldn5, and Cldn7 in the enteroid monolayers derived from *Slc39a8*-IEC KO mice. A similar trend was observed for the Cldn3 protein levels, although the difference did not reach statistical significance. We have added these new data to the manuscript as **Fig. 7e** and indicated the changes in the text and figure legends.

7. *Will ACER1 inhibitor rescue the IL-10 KO mice?*

We intended to do this experiment; however, for unknown reasons, IL-10 KO mice do not develop spontaneous colitis in our animal facility. Consequently, we were unable to directly test the effect of the ACER1 inhibitor in rescuing colitis in IL-10 KO mice. However, based on our findings that ACER1 is upregulated in the intestines of *Slc39a8*-IEC KO mice and that ACER1 inhibitor mitigates colitis in these mice, we speculate that if IL-10 KO mice had displayed a deficiency in *Slc39a8* and Mn levels resulting in up-regulation of ACER1, that treatment with an ACER1 inhibitor could potentially attenuate colitis in IL-10 KO mice. Further studies would be required to investigate this hypothesis.

8. *More discussion should be added on human IBD if the authors believe that Mn deficiency up-regulates Acer1 expression in the DSS-colitis model.*

Our findings in mice align with epidemiological studies that have reported a positive relationship between Mn deficiency and the risk of IBD. For instance, a recent survey of pediatric patients newly diagnosed with Crohn's disease and ulcerative colitis showed significantly lower Mn levels in their hair samples⁵. Another human study reported that blood Mn levels are significantly lower in patients with active Crohn's disease and ulcerative colitis than in patients in remission⁶. These findings suggest that IBD patients may be at risk of Mn deficiency, leading to potential up-regulation of *Acer1* expression. Therefore, interventions aimed at restoring ACER1 levels may hold promise for improving human IBD conditions. We have further elaborated on this point in the **Discussion** section, as indicated by the yellow-highlighted text.

9. *A working model of ACER1, Mn deficiency, and barrier in the development of chronic intestinal inflammation will help the readers.*

We appreciate the reviewer's suggestion and have included a working model in our revised manuscript to provide readers with a visual summary of the proposed mechanism involving ACER1, Mn deficiency, and barrier function in the development of chronic intestinal inflammation (**Fig. 9i**).

Reviewer #2

Choi et al. have submitted a solid manuscript that interrogates the role of the manganese transporter *Slc39a8* in intestinal manganese absorption and inflammatory bowel disease using cell culture and mouse models. The work is highly original and will have a prominent impact on the field of manganese biology and intestinal disease. We do have several recommendations on how the manuscript can be improved. These are described here:

We appreciate the positive feedback from the reviewer regarding our manuscript.

1. We highly recommended separating males and females for each panel instead of pooling them together. Even though no sex-specific differences are observed for Mn levels in Fig. 1, this doesn't necessarily mean such differences do not occur for other phenotypes. For instance, in Fig. S4, the authors show that female mice have lower sensitivity to effects of DSS.

We have now included separate data for both male and female mice for *Slc39a8* expression (Fig.1a and 1b), tissue Mn levels (Fig. 1d), and Mn radiotracer assays (Fig. 2). In these measurements, we did not find significant sex differences. Thus, the lower sensitivity of female mice to DSS-induced colitis is not caused by differences in *Slc39a8* expression or Mn levels between sexes. Consequently, for further analysis, we focused on male mice, which resulted in the inability to separate males and females in other panels.

2. Additionally, the authors point out in the discussion that “inconsistent Mn concentrations among the two studies (#39,40) [of *Slc39a8* A391T KI mice] complicate any interpretations of how *SLC39A8* deficiency alters Mn homeostasis and exacerbates DSS-induced colitis”—could sex-specific differences contribute to the inconsistencies in these studies?

We appreciate the reviewer's question. As discussed in the manuscript, two recent studies independently reported altered Mn homeostasis and exacerbated DSS-induced colitis in *Slc39a8* A391T KI mice^{7,8}. Please see the summary table below comparing tissue Mn levels in *Slc39a8* A391T knock-in (KI) mice from two studies. Sunuwar et al.⁵ reported reduced Mn levels in whole blood, liver, and kidney of *Slc39a8* A391T KI mice but no alterations in other tissues, including the distal small intestine, colon, lung, heart, and brain. Nakata et al.⁸ reported reduced Mn levels in whole blood, liver, and colon of *Slc39a8* A391T KI mice, but they did not examine Mn levels in other tissues⁸. The colon Mn data between the two studies were inconsistent. However, whole blood is the only cell type for which the two reports measured Mn levels in both sexes; Mn in other tissues was only measured in male animals. In whole blood, no sex difference was detected. Therefore, it is difficult to conclude whether the inconsistent colon Mn data reflect the sexes examined in the two studies. We have edited the **Discussion** section accordingly in the yellow-highlighted text.

Table. Tissue Mn Levels of *Slc39a8* A391T KI mice reported in the literature.

		Sunuwar et al. ⁵		Nakata et al. ⁸	
		Male	Female	Male	Female
Tissue Mn Levels	Whole blood	Reduced Mn levels	Reduced Mn levels	Reduced Mn levels	Reduced Mn levels
	Distal small intestine	No alterations in Mn levels	-	-	-

	Colon	No alterations in Mn levels	-	Reduced Mn levels	-
	Liver	Reduced Mn levels	-	Reduced Mn levels	-
	Kidney	Reduced Mn levels	-	-	-
	Lung	No alterations in Mn levels	-	-	-
	Heart	No alterations in Mn levels	-	-	-
	Brain	No alterations in Mn levels	-	-	-

3. *This study used a mixture of three genotypes as a control (Slc39a8-flox (Slc39a8fl/fl and Slc39a8fl/+), Slc39a8-WT (Slc39a8+/+), and Vil-Cre (Slc39a8+/+:Villin-Cre)). Authors should not use four different genotypes as controls. This is a key issue that needs to be addressed.*

This is an important point. We took great care to compare key experimental results, such as metal concentrations and experimentally induced colitis models, among all four control genotypes. We found no significant differences in Mn concentrations in the ileum and colon among the four control genotypes at 6–8 weeks of age (**Supplementary Fig. 9a**). We also observed that the four genotypes of controls did not display significant differences in DSS sensitivity at 6–8 weeks of age (**Supplementary Fig. 9b**). These data indicate that the flox sequence and Cre transgene did not impact Mn homeostasis or DSS sensitivity. We have added this information to **Supplementary Fig. 9a and 9b**, and the *Animals* in the **Methods** section.

4. *For the ⁵⁴Mn gavage experiments, the authors collected blood 15 minutes after gavage and tissues four hours after gavage. Four hours is more than enough time for ⁵⁴Mn to have undergo biliary excretion and enterohepatic circulation. Can the authors comment on why tissues weren't harvested at the same time as blood and how this discrepancy would influence their results?*

We appreciate the reviewer's comment. We chose the two different time points for assaying the blood and other organs based on prior studies in which we monitored the relatively fast entry of Mn into blood circulation compared to the entry into tissues^{9, 10}. The observed ⁵⁴Mn levels in the tissues after a 4 h interval from gavage or intravenous injection may reflect enhanced Mn excretion. However, we found a significant reduction in transcript levels of intestinal Mn excretion proteins, *Slc39a14/ZIP14* and *Slc30a10/ZnT10*, in *Slc39a8*-IEC KO-derived enteroids (**Supplementary Fig. 3a**), thereby limiting the possibility of enhanced excretion of Mn, at least in the intestine. Future studies are warranted to determine the contribution of absorption and excretion of Mn in each tissue. We have added this information to the **Results** section and in the yellow-highlighted text.

5. *It appears that the confocal immunofluorescence images in Fig. 3b are misaligned—ZO-1 staining crosses through nuclei!*

We appreciate the reviewer's careful observation. We have provided new confocal immunofluorescence images in **Fig. 3b** to ensure an accurate representation of our data.

6. *Were there any changes in expression of Dmt1 or other relevant metal transporters in the intestines of the Slc39a8 IEC KO mice?*

We did provide the data on these changes in **Supplementary Fig. 3a**, in the original manuscript, although this may not have been easily recognizable. Our results showed that the *Slc39a8*-IEC KO-derived enteroids exhibited significantly increased *Slc11a2/DMT1* (divalent metal transporter 1) transcript levels, along with significantly decreased *Slc39a14/ZIP14*, *Slc30a10/ZnT10*, and *Slc40a1/FPN* (ferroportin-1) transcript levels. These findings suggest that DMT1 may compensate for the loss of SLC39A8 expression in the intestines of the *Slc39a8*-IEC KO mice.

7. *Ceramide accumulation is known to mediate inflammation (<https://www.nature.com/articles/nm1748>). ACER1 inhibition will result in ceramide accumulation according to the pathway shown in Fig. 6c, so would ACER1 inhibition be expected to exacerbate colitis? (Also, alkaline ceramidase 3 deficiency aggravates colitis and colitis-associated tumorigenesis in mice by hyper-activating the innate immune system <https://www.nature.com/articles/cddis201636>). Please comment on this.*

Thank you for bringing up this important point. Given the similarity in content between your questions 7 and 8, I have provided a consolidated response that addresses the common points raised in both questions, thereby avoiding redundancy and ensuring clarity in the feedback provided.

To determine the mechanisms underlying the upregulation of ACER1 and the rescue effect of D-e-MAPP, we performed global, unbiased lipidomics profiling of the intestine from WT and mutant mice with or without the inhibitor using triple time of flight liquid chromatography-mass spectrometry (Triple-TOF LC-MS). While ACER1 converts ceramide into sphingosine, sphingomyelin synthase (SMS1) converts ceramide into sphingomyelin; therefore, these two enzymes represent the two major arms of ceramide metabolism (**Fig. 8j**). The lipidome profiling identified 40 differentially regulated lipids in *Slc39a8*-IEC KO intestines compared to control ($P_{adj} < 0.2$, **Supplementary Fig. 7a**). Among the altered lipids, the heatmap (**Fig. 8k**) represents lipid species relevant to ceramide metabolism, including sphingomyelins, hexosylceramides, sphinganine, phosphatidylethanolamine ceramides, and ceramides. While we predicted a reduction in ceramides following ACER1 upregulation, the reduction was only observed for Cer 38:2;2O and Cer 43:1;2O, whereas other four ceramide species were upregulated in the *Slc39a8*-IEC KO intestines (**Supplementary Fig. 7b-g**).

Notably, sphingomyelins were all significantly downregulated in the mutant (**Fig. 8m**), which is consistent with the fact that Mn is necessary for the activity of ceramide phosphoethanolamine synthase (Cpes)¹¹, the insect homolog of SMS1¹². Indeed, SMS activity was lower in *Slc39a8*-IEC KO intestines compared to control intestines (**Fig. 8l**). Furthermore, all three sphingomyelin species showed partial or complete restoration of their abundance upon D-e-MAPP treatment; the inhibitor alone did not affect the sphingomyelin levels (**Fig. 8m**). By contrast, we did not find any ceramide species whose changes completely agreed with the phenotypic outcomes; i.e., dysregulation in *Slc39a8*-IEC KO, restoration by D-e-MAPP and no change by inhibitor alone (**Supplementary Fig. 7b-g**); the six KO-dysregulated ceramides largely satisfied the first two criteria, but D-e-MAPP alone also changed their levels in the control intestine, ruling them out as the mediator for gut phenotype rescue effects.

These results led us to propose the following mechanisms underlying Acer1 upregulation in *Slc39a8*-IEC KO intestines and the phenotypic rescue by D-e-MAPP. First, the reduced SMS1 activity due to Mn deficiency led to reduced sphingomyelin and increased ceramide levels, thereby triggering the compensatory increase of Acer1 expression to offset the ceramide increase. The increased Acer1

expression indirectly led to alterations in lipid composition, including the upregulation of some ceramide species and dysregulation of other detected lipids. The phenotypic rescues by D-e-MAPP (**Figs. 7, 8, and 9** and **Supplementary Fig. 6**) suggest that the upregulation of *Acer1* and associated lipidome alterations contribute to the impaired gut function in *Slc39a8*-IEC KO mice. However, the rescue effect is unlikely to be mediated by ceramides; instead, the restoration of other lipid species, such as sphingomyelin, is the candidate mediator for the observed impact of D-e-MAPP on gut physiology. We have incorporated these new data into the manuscript as **Fig. 8j, 8k, 8l, and 8m** and **Supplementary Fig. 7a, 7b, 7c, 7d, 7e, 7f, and 7g**, and have indicated the corresponding changes in the text and figure legends.

Regarding ACER3, we conducted additional experiments to measure the mRNA levels of the other ceramidases, including acid ceramidase (*Asah1*), neutral ceramidase (*Asah2*), and alkaline ceramidases 1, 2, and 3 (*Acer1*, *Acer2*, and *Acer3*). Our qPCR analysis revealed no significant differences in the mRNA levels of *Asah1*, *Asah2*, *Acer2*, and *Acer3* in the ileal and colonic mucosa from *Slc39a8*-IEC KO mice and control mice. However, we observed a significant upregulation of *Acer1* expression in the ileal and colonic mucosa from *Slc39a8*-IEC KO mice compared to control mice, which is consistent with our findings shown in **Fig. 6c and 6d**. These new results have been included in the **Supplementary Fig. 8**, and the relevant information has been incorporated into the text and Supplemental Figure legends.

8. *Did the authors measure ceramide and sphingosine levels in cell culture and mouse models with and without Acer1 inhibition? This would help to confirm the physiologic relevance of altered Acer1 expression and the efficacy of Acer1 inhibition.*

Please see the above response to your question 7. Regarding the lipid levels in the cell cultures, we indeed attempted the lipid measurements; however, we were not able to measure the lipid levels reliably due to the low amounts of materials.

9. *The authors state that the increased spleen mass in Slc39a8 IEC KO mice implicates splenic macrophage infiltration. Was splenic macrophage infiltration observed by histology?*

To address this, we performed H&E staining of the spleen tissues from both control and *Slc39a8* IEC KO mice. Indeed, our histological analysis revealed increased follicle destruction and infiltration of inflammatory cells in the spleens of *Slc39a8* IEC KO mice compared to the control, as demonstrated in the newly added data in **Supplementary Fig. 3c** and indicated in the text and Supplementary Figure legends.

10. *The histological images included in figures are quite small. The authors are recommended to include images of the same magnification and large enough to visualize differences between images.*

We appreciate the reviewer's feedback. We have made efforts to ensure that the images are clear and representative of the observed differences between the groups (**Fig. 4f, Fig. 5e, Fig. 8e, Fig. 9g, and Supplementary Fig. 4d**). We also revised the figure legends to clearly indicate the magnification of the images for better understanding.

11. *It is striking that only four genes were found to be differentially expressed in RNA seq analysis. Can the authors confirm that this is correct? Also, do the genes other than Acer1 have potential implications for the study at hand? What is known about these other genes?*

These issues need to be addressed.

Yes, only four genes reached our statistical threshold ($P_{adj} < 0.1$) with DEseq2 analysis. We now include the full results for the DEseq2 analysis as **Supplementary Material**. Among these four genes, *Slc39a8* was the only downregulated gene in the *Slc39a8*-IEC KO intestines. The three upregulated genes are *Ighv1-55* (ENSMUSG00000095589), *Entpd4b*, and *Acer1*.

Ighv1-55 has no known function, while *Entpd4b* encodes ectonucleoside triphosphate diphosphohydrolase 4 (ENTPD4), a member of the apyrase protein family involved in the hydrolysis of nucleotide diphosphates and triphosphates¹³. *Acer1* encodes alkaline ceramidase 1 (ACER1), which catalyzes the hydrolysis of very long chain ceramides to generate sphingosine^{14, 15}. Given the significant association between sphingolipid metabolism and IBD¹⁶, we selected to further investigate ACER1 in our study.

We have included a new table in the manuscript (**Supplemental Table 1**) that provides detailed information on these four genes, including gene description, p -value, direction of misregulation, known function, and references. This information is also incorporated in the revised **Discussion** section, with the relevant text highlighted in yellow.

Supplemental Table 1. The four most dysregulated genes in *Slc39a8*-IEC KO intestines.

Genes	Gene Description	Padj	Known Function
Downregulated			
Slc39a8	Solute Carrier Family 39 Member 8	1.25×10^{-5}	Cellular import of divalent metal ions ^{17, 18, 19, 20}
Upregulated			
Ighv1-55 (ENSMUSG00000095589)	Immunoglobulin Heavy Variable 1-55	3.40×10^{-4}	Unknown function
Entpd4b	Ectonucleoside Triphosphate Diphosphohydrolase 4	2.56×10^{-2}	Hydrolysis of nucleotide diphosphates and triphosphates ¹³
Acer1	Alkaline Ceramidase 1	4.65×10^{-2}	Hydrolysis of very long chain ceramides to generate sphingosine ^{14, 15}

12. *It appears that a legend is missing from Fig. 9B.*

We have added a legend to **Fig. 9b**, and the corresponding text has been highlighted in yellow.

13. *The authors are recommended to include a model figure summarizing their results.*

We have included a model figure (**Fig. 9i**) in our revised manuscript to summarize our results concisely.

14. *The authors suggest that Mn deficiency reduces Cpes activity in the discussion about how Slc39a8 IEC KO mice develop misregulated Acer1 gene expression. Can the authors*

measure Cpes activity in their cell culture and mouse models?

We conducted additional experiments to measure the activity of sphingomyelin synthase 1 (SMS1), which is the mammalian homolog of ceramide phosphoethanolamine synthase (Cpes). We measured SMS1 activity in both cell culture and mouse models, and we observed a significant reduction in SMS1 activity in the intestines of *Slc39a8*-IEC KO mice and *Slc39a8*-IEC KO-derived intestinal organoid monolayer cells (**Fig. 8I**). These findings are in line with previous studies in insects, which have shown that Mn is required for the enzyme activity of ceramide phosphoethanolamine synthase (Cpes)¹¹, the insect homolog of SMS1¹². We appreciate the reviewer's suggestion to investigate this aspect, as it has enhanced the comprehensiveness of our study.

Reviewer #3

The authors present data demonstrating a role for SLC39A8 in manganese (Mn) absorption. Intestinal specific deletion of SLC398A resulted in decreased Mn in tissues and circulation and rendered mice more susceptible to intestinal injury with dextran sulfate sodium. This loss also lead to an increase in ACER1 mRNA in intestinal tissues. The authors then demonstrate D-e-MAPP (an alkaline ceramidase inhibitor) partially protects from DSS-induced intestinal injury. The work is well done and well controlled; however, a critical supplemental experiment is missing and some critical points are missing from the discussion.

We thank the reviewer for these supportive comments.

1. *The authors should check the mRNA levels of the other ceramidases (there are 5 total - 1 neutral, 1 acid, and 3 alkaline).*

To address this point, we conducted additional experiments to measure the mRNA levels of the other ceramidases, including acid ceramidase (*Asah1*), neutral ceramidase (*Asah2*), and alkaline ceramidases 1, 2, and 3 (*Acer1*, *Acer2*, and *Acer3*). Our qPCR analysis revealed no significant differences in the mRNA levels of *Asah1*, *Asah2*, *Acer2*, and *Acer3* in the ileal and colonic mucosa from *Slc39a8*-IEC KO mice and control mice (**Supplementary Fig. 8**). However, we observed a significant upregulation of *Acer1* expression in the ileal and colonic mucosa from *Slc39a8*-IEC KO mice compared to control mice, which is consistent with our findings shown in **Fig. 6c and 6d**. These new results have been included in **Supplementary Fig. 8**, and the relevant information has been incorporated into the text and Supplemental Figure legends.

2. *D-e-MAPP inhibits all 3 alkaline ceramidases. This is one of the critical points missing in the discussion. It is a specific alkaline ceramidase inhibitor at the doses used, but inhibits more than ACER1.*

We agree that ACER2 and ACER3 inhibition by D-e-MAPP could mediate the phenotypic alterations in *Slc39a8*-IEC KO mice. A more plausible scenario is that ACER1 mediates the impact of D-e-MAPP because we did not detect any discernible expression changes in ACER2 or ACER3 expression in *Slc39a8*-IEC KO intestines (**Supplementary Fig. 8**). However, the lack of expression changes still does not rule out the possible involvement of *Acer2* and *Acer3*. We explored this aspect in the **Discussion** section.

3. *There are also studies using alkaline ceramidase 3 (ACER3) deficient mice in colitis and colitis associated cancer. This is not included in the discussion, but likely should be.*

Thank you for your valuable input. We have added the following paragraph to the **Discussion** section as suggested.

In addition to Acer1, two other members of the alkaline ceramidase family, Acer2 and Acer3, are present in both mice and humans^{15, 21}. D-e-MAPP can inhibit ACER2 and ACER3^{22, 23}; however, since we did not find any expression changes in Acer2 and Acer3 in *Slc39a8*-IEC KO mice (**Supplementary Fig. 8**), the rescue effect observed with D-e-MAPP was likely due to ACER1 inhibition. Nevertheless, the lack of expression changes still does not rule out a potential involvement of Acer2 and Acer3. A previous study has demonstrated that ACER3 plays a crucial role in mediating the immune response in innate immune cells and colonic epithelial cells²⁴. Acer3 deficiency has been shown to increase the local and systemic production of proinflammatory cytokines and exacerbated colitis in the DSS-induced murine colitis model²⁴. These observations indicate that Acer activities are tightly regulated to ensure normal gut functions.

References

1. Weber CR, Nalle SC, Tretiakova M, Rubin DT, Turner JR. Claudin-1 and claudin-2 expression is elevated in inflammatory bowel disease and may contribute to early neoplastic transformation. *Lab Invest* **88**, 1110-1120 (2008).
2. Zeissig S, *et al.* Changes in expression and distribution of claudin 2, 5 and 8 lead to discontinuous tight junctions and barrier dysfunction in active Crohn's disease. *Gut* **56**, 61-72 (2007).
3. Dong L, *et al.* Mannose ameliorates experimental colitis by protecting intestinal barrier integrity. *Nat Commun* **13**, 4804 (2022).
4. Mayangsari Y, Suzuki T. Resveratrol Ameliorates Intestinal Barrier Defects and Inflammation in Colitic Mice and Intestinal Cells. *J Agric Food Chem* **66**, 12666-12674 (2018).
5. Cho JM, Yang HR. Hair Mineral and Trace Element Contents as Reliable Markers of Nutritional Status Compared to Serum Levels of These Elements in Children Newly Diagnosed with Inflammatory Bowel Disease. *Biol Trace Elem Res* **185**, 20-29 (2018).
6. Silva A SM, Amarante H. Food intake in patients with inflammatory bowel disease. *Arq Bras Cir Dig* **24**, 204-209 (2011).
7. Sunuwar L, *et al.* Pleiotropic ZIP8 A391T implicates abnormal manganese homeostasis in complex human disease. *JCI Insight* **5**, (2020).

8. T N, *et al.* A missense variant in SLC39A8 confers risk for Crohn's disease by disrupting manganese homeostasis and intestinal barrier integrity. *Proceedings of the National Academy of Sciences of the United States of America* **117**, (2020).
9. Seo YA, Wessling-Resnick M. Ferroportin deficiency impairs manganese metabolism in flatiron mice. *Faseb j* **29**, 2726-2733 (2015).
10. A S, *et al.* Intestinal DMT1 is critical for iron absorption in the mouse but is not required for the absorption of copper or manganese. *American journal of physiology Gastrointestinal and liver physiology* **309**, (2015).
11. Vacaru AM, van den Dikkenberg J, Ternes P, Holthuis JC. Ceramide phosphoethanolamine biosynthesis in *Drosophila* is mediated by a unique ethanolamine phosphotransferase in the Golgi lumen. *J Biol Chem* **288**, 11520-11530 (2013).
12. Panevska A, Skočaj M, Križaj I, Maček P, Sepčić K. Ceramide phosphoethanolamine, an enigmatic cellular membrane sphingolipid. *Biochim Biophys Acta Biomembr* **1861**, 1284-1292 (2019).
13. Biederbick A, Rose S, Elsässer HP. A human intracellular apyrase-like protein, LALP70, localizes to lysosomal/autophagic vacuoles. *J Cell Sci* **112 (Pt 15)**, 2473-2484 (1999).
14. Houben E, *et al.* Differentiation-associated expression of ceramidase isoforms in cultured keratinocytes and epidermis. *J Lipid Res* **47**, 1063-1070 (2006).
15. Sun W, *et al.* Upregulation of the human alkaline ceramidase 1 and acid ceramidase mediates calcium-induced differentiation of epidermal keratinocytes. *J Invest Dermatol* **128**, 389-397 (2008).
16. Abdel Hadi L, Di Vito C, Riboni L. Fostering Inflammatory Bowel Disease: Sphingolipid Strategies to Join Forces. *Mediators Inflamm* **2016**, 3827684 (2016).
17. NA B, M K, Y M, M M, K T, T S. Mycobacterium bovis BCG cell wall and lipopolysaccharide induce a novel gene, BIGM103, encoding a 7-TM protein: identification of a new protein family having Zn-transporter and Zn-metalloprotease signatures. *Genomics* **80**, (2002).
18. TP D, *et al.* Identification of mouse SLC39A8 as the transporter responsible for cadmium-induced toxicity in the testis. *Proceedings of the National Academy of Sciences of the United States of America* **102**, (2005).
19. L H, *et al.* ZIP8, member of the solute-carrier-39 (SLC39) metal-transporter family: characterization of transporter properties. *Molecular pharmacology* **70**, (2006).

20. Wang CY, *et al.* ZIP8 is an iron and zinc transporter whose cell-surface expression is up-regulated by cellular iron loading. *J Biol Chem* **287**, 34032-34043 (2012).
21. Xu R, *et al.* Golgi alkaline ceramidase regulates cell proliferation and survival by controlling levels of sphingosine and S1P. *Faseb j* **20**, 1813-1825 (2006).
22. Sugita M, Williams M, Dulaney JT, Moser HW. Ceramidase and ceramide synthesis in human kidney and cerebellum. Description of a new alkaline ceramidase. *Biochim Biophys Acta* **398**, 125-131 (1975).
23. Bielawska A, *et al.* (1S,2R)-D-erythro-2-(N-myristoylamino)-1-phenyl-1-propanol as an inhibitor of ceramidase. *J Biol Chem* **271**, 12646-12654 (1996).
24. Wang K, *et al.* Alkaline ceramidase 3 deficiency aggravates colitis and colitis-associated tumorigenesis in mice by hyperactivating the innate immune system. *Cell Death Dis* **7**, e2124 (2016).

REVIEWER COMMENTS

Reviewer #1 (Remarks to the Author):

The authors have addresses this reviewer's concerns in their revision.

Reviewer #2 (Remarks to the Author):

The authors made substantial efforts to address the concerns raised in the initial round of reviews. We do have two additional comments:

1) We appreciate the effort made by the authors to address sex-specific differences in phenotypes. However, in Fig 2b (and others) where authors separated sex of mice, it is not prudent to run statistics on data with N=2. Ideally authors should have included three or more mice per sex.

2) Please include full uncropped blots for immunoblots as supplemental data.

Reviewer #3 (Remarks to the Author):

The authors have significantly improved the quality of the manuscript and included several new and necessary experiments. However, their untargeted lipidomic studies are insufficient to draw the conclusions outlined in the manuscript. Untargeted lipidomics provides compositional data, not quantitative assessments of lipid levels. Several of the SM and Ceramide species that were identified in the untargeted method are not "canonical" sphingolipids and therefore may have been mis-identified. Moreover, untargeted lipidomics is often not sensitive enough to measure sphingosine - the produce of ACER1. Targeted lipidomic measurements for SM, CER, sphingoid bases and HexosylCer would significantly improve the validity of the conclusions drawn from the data. In addition, ACER1 not only produces sphingosine, but SPH is the substrate for sphingosine kinases (SKs), which have also been extensively implicated in IBD. This should be added to the discussion and potentially mRNA levels/protein levels of SKs added to the data.

Response to reviewers

We thank the reviewers for the thoughtful comments and suggestions. As detailed in a point-by-point response below, we have addressed all of them with additional experiments, and we feel that the manuscript has been substantially strengthened as a result.

Reviewer #1:

The authors have addresses this reviewer's concerns in their revision.

We sincerely appreciate the reviewer's positive feedback.

Reviewer #2:

The authors made substantial efforts to address the concerns raised in the initial round of reviews. We do have two additional comments:

We sincerely appreciate the reviewer's positive feedback. We also thank the reviewer's comments for further improving our manuscript.

1. *We appreciate the effort made by the authors to address sex-specific differences in phenotypes. However, in Fig 2b (and others) where authors separated sex of mice, it is not prudent to run statistics on data with N=2. Ideally authors should have included three or more mice per sex.*

Because we do not have enough N in Fig. 2b, we decided not to present sex-separated statistical analysis. Investigating of the sex differences in intestinal absorption of ⁵⁴Mn is a potential subject for future research.

2. *Please include full uncropped blots for immunoblots as supplemental data.*

In compliance with the journal's requirements, we have included the full, uncropped immunoblots in the **source data** file, not in supplemental data.

Reviewer #3:

The authors have significantly improved the quality of the manuscript and included several new and necessary experiments.

1. *However, their untargeted lipidomic studies are insufficient to draw the conclusions outlined in the manuscript. Untargeted lipidomics provides compositional data, not quantitative assessments of lipid levels. Several of the SM and Ceramide species that were identified in the untargeted method are not "canonical" sphingolipids and therefore may have been mis-identified. Moreover, untargeted lipidomics is often not sensitive enough to measure sphingosine - the produce of ACER1. Targeted lipidomic measurements for SM, CER, sphingoid bases and HexosylCer would significantly improve the validity of the conclusions drawn from the data.*

We greatly appreciate your thoughtful comments and the opportunity to clarify our approach to untargeted lipidomics in response to your concerns. We would like to address each of your points as follows:

In our current manuscript, lipids were identified using the MS/MS product ion fragmentation of molecular ions (m/z range 200-1200) resolved by RP-UPLC-ESI-QTOF-MS/MS. The method is highly sensitive and reproducible even for profiling low abundance lipids. We acknowledge the reviewer's concern that compound identification in untargeted lipidomics methods can be prone to error when MS/MS search results alone are used and trusted without further review. However, we do not find that identification in untargeted lipidomics is inherently flawed when an appropriate review of identifications is undertaken.

First, regarding the MS/MS spectral search strategy for initial compound identification, we used the "LipidBlast" library coupled with National Institute of Standards and Technology (NIST) spectral search tools, which matches compounds by observed mass and MS/MS fragmentation pattern. To facilitate accurate lipid identification, a more stringent mass error rate of 0.001 m/z for the positive mode and 0.005 m/z for the negative mode was determined based on the mass accuracy of internal standards to reduce the rate of false positives and identify likely correct candidates. The sphingolipid species considered for identification were limited to features with (%RSD < 20) in pooled samples. More details regarding the spectral matching procedure used for compound identification against the LipidBlast library are outlined in the original report¹.

A manual review of spectral matches was performed in all potentially uncertain cases, including the sphingolipid species questioned by the reviewers. Examples of spectral matches are now included in **Supplementary Figures 10a and 10b**. The product ion scan of $[M+Na]^+$ species of sphingomyelin (SM 34:1, 2O) yields three fragments corresponding to NL of C₃H₉N (m/z 666.477, C₃₆H₇₀NO₆PNa⁺), NL of C₅H₁₄NO₄P (m/z 542.486, C₃₄H₆₅NO₂Na⁺), NL of C₅H₁₆NO₅P (m/z 502.497, C₃₄H₆₄NO⁺), respectively. A product ion scan of m/z 300.2860, which corresponds to the $[M+H]^+$ ion for SPB 18:1;2O molecular species, was searched against the LipidBlast In-Silico library, revealing an exact match for the expected compound with high library scores (**Supplementary Figure 10a**). MS/MS spectra of ceramides in the positive mode were dominated by fragments from the sphingoid base (**Supplementary Figure 10b**). Collisional activation of ceramides in positive ion mode showed the characteristic product ions of m/z 264.2684 [C₁₈H₃₄N]⁺ and 282.279 that were attributed to the loss of N-linked fatty acid and one and two water molecules, respectively. We also added the following data, according to the latest communication with the reviewer #3. The positive mode MSMS product ion spectrum of HexCer 18:0;3O/16:0;(2OH), which is likely one of the non-canonical ceramides questioned by the reviewer, is also illustrated in **Supplementary Figure 10b**. This compound shows an NL of C₆H₁₀O₅, the hexose ring, results m/z 572.5232 (**Supplementary Figure 10b**), as well as other peaks that are m/z 554.5143 and m/z 264.2686 correspond to NL of C₆H₁₀O₅ and H₂O and SPB 18:0;O₃ -3H₂O respectively.

To provide additional evidence for the accuracy of our compound identifications, chromatographic retention time can be used in lipidomics performed with reversed-phase liquid chromatography. Lipids of a particular class (sphingolipids in this case) can be expected to follow a regular order of elution, wherein chromatographic retention time should increase with increasing alkyl chain length and decrease with increasing degree of desaturation (number of double bonds). Deviations from the observed elution pattern can highlight potentially misidentified compounds. To investigate these patterns with the sphingolipid species in question, we plotted sphingolipid RT against alkyl chain length and highlighted degree of desaturation using different color marker points. As seen in the scatter plot (**Supplementary Figures 10c and 10d**), observed retention time increases with increasing alkyl chain length, and RT decreases with increasing number of double bonds. All such plots follow a highly linear pattern for a given compound class and show no substantial RT deviation or any of the detected species, supporting the accuracy of our identifications.

Last, the reviewers raised the question regarding the validity of the quantitation of untargeted lipidomics data compared to targeted methods. We acknowledge that true absolute quantitation of lipids should only be performed using a targeted strategy with authentic standards and compound-matching isotopically labeled internal standards. However, relative quantitative measurements are certainly valid from untargeted lipidomics methods, as has been demonstrated previously².

Overall, it is our conclusion that with appropriate controls and a review of compound identification and method performance, the identification and relative quantitative analysis of lipids using a carefully managed untargeted workflow is not necessarily more prone to error than targeted methods. Each technique still has its advantages and disadvantages; targeted methods can achieve greater sensitivity under certain circumstances than untargeted ones, whereas untargeted ones enable the detection of un-predicted lipid species not included in a multiplexed selected reaction monitoring method.

Taken together, we included **Supplementary Figures 10a-d** in the revised version of the manuscript and incorporate the relevant information into the **Methods** section.

2. *In addition, ACER1 not only produces sphingosine, but SPH is the substrate for sphingosine kinases (SKs), which have also been extensively implicated in IBD. This should be added to the discussion and potentially mRNA levels/protein levels of SKs added to the data.*

To address this point, we conducted additional experiments to assess the mRNA levels of sphingosine kinases (SphKs), specifically SphK1 and SphK2. Our qPCR analysis revealed a significant upregulation of *Sphk1* in the ileum and colon of *Slc39a8*-IEC KO compared to control group (**Supplementary Fig. 8f**). However, no significant differences were observed in the mRNA levels of *Sphk2* in the ileum and colon of both control and *Slc39a8*-IEC KO mice (**Supplementary Fig. 8g**).

We also conducted immunoblot analysis targeting SphK1 and SphK2 in the ileum and colon of both control and *Slc39a8*-IEC KO mice. We used two different antibodies for each protein—anti-SphK1 antibodies (Thermo Fisher, Cat. No. PA514068 and Abclonal, Cat. No. A0139) and anti-SphK2 antibody (Thermo Fisher, Cat. No. PA551064 and Abclonal, Cat. No. A6748). However, we were unable to detect SphK1 and SphK2 proteins at the predicted molecular weights from intestine tissues of both control and *Slc39a8*-IEC KO mice. The updated results of mRNA expression levels of *Sphk1* and *Sphk2* have been included in **Supplementary Fig. 8f and 8g**, and the relevant information has been incorporated into the revised text and Supplemental Figure legends.

As suggested, we have incorporated the following point into the **Discussion** accordingly. In addition to ACER1's role in sphingosine production, sphingosine serves as the substrate for sphingosine kinases (SphKs), extensively implicated in IBD. Three ceramidases, operating in distinct cellular compartments, convert ceramide into sphingosine, which is then phosphorylated by either sphingosine kinase 1 (Sphk1) or sphingosine kinase 2 (Sphk2) to form sphingosine-1-phosphate (S1P). Studies in the mouse small intestine have shown twofold higher SphK activity compared to the colon, with SphK1 being the predominant isoform in the colon³. In a rat model of intestinal inflammation, the 1-phosphorylated forms of ceramide (C1P) and sphingosine (S1P) increase with inflammation⁴. Elevated S1P levels are particularly relevant in IBD, observed not only in the intestine but also in the blood due to increased SphK1 expression⁵. Conversely, SphK2 deficiency has been shown to reduce the severity of IBD⁶. These studies collectively suggest the involvement of the SphK/S1P pathway in the development of IBD. In line with these observations, our *Slc39a8*-IEC KO mice exhibited increased *Sphk1* mRNA expression in the ileum and colon, whereas *Sphk2* expression levels remain unchanged (**Supplementary Figure 8f and 8g**), indicating a potential upregulation of the SphK1/S1P pathway. However, our lipidomics data indicate no changes in

sphingosine, likely due to compensatory mechanisms. Future studies are needed to explore the intricate regulation of sphingosine metabolism in response to *Slc39a8* deficiency.

We would like to thank our reviewers for all the thoughtful comments and suggestions, and we believe that these revisions have significantly improved the manuscript.

Sincerely yours,
Young-Ah Seo

Reference

1. Kind T, Liu KH, Lee DY, DeFelice B, Meissen JK, Fiehn O. LipidBlast in silico tandem mass spectrometry database for lipid identification. *Nat Methods* **10**, 755-758 (2013).
2. Cajka T, Smilowitz JT, Fiehn O. Validating Quantitative Untargeted Lipidomics Across Nine Liquid Chromatography-High-Resolution Mass Spectrometry Platforms. *Anal Chem* **89**, 12360-12368 (2017).
3. Fukuda Y, Kihara A, Igarashi Y. Distribution of sphingosine kinase activity in mouse tissues: contribution of SPHK1. *Biochem Biophys Res Commun* **309**, 155-160 (2003).
4. Dragusin M, *et al.* Effects of sphingosine-1-phosphate and ceramide-1-phosphate on rat intestinal smooth muscle cells: implications for postoperative ileus. *Faseb j* **20**, 1930-1932 (2006).
5. Snider AJ, *et al.* A role for sphingosine kinase 1 in dextran sulfate sodium-induced colitis. *Faseb j* **23**, 143-152 (2009).
6. Samy ET, *et al.* Cutting edge: Modulation of intestinal autoimmunity and IL-2 signaling by sphingosine kinase 2 independent of sphingosine 1-phosphate. *J Immunol* **179**, 5644-5648 (2007).

REVIEWERS' COMMENTS

Reviewer #2 (Remarks to the Author):

All issues have been addressed.

Reviewer #3 (Remarks to the Author):

The authors have addressed my concerns.